# New insight into the formation and aging processes of organic aerosol from positive matrix factorization (PMF) analysis of ambient FIGAERO-CIMS thermograms

Mingfu Cai[1,2,3], Bin Yuan[2,3], Weiwei Hu[4,5*], Ye Chenshuo[6], Shan Huang[2,3], Suxia Yang[7], Wei Chen[4,], Yuwen Peng[2,3], Zhaoxiong Deng[2,3], Jun Zhao[8†], Duohong Chen[9], Jiaren Sun[1], Min Shao[2,3]

[1]Guangdong Province Engineering Laboratory for Air Pollution Control, Guangdong Provincial Key Laboratory of Water and Air Pollution Control, South China Institute of Environmental Sciences, MEE, Guangzhou 510655, China

[2]Institute for Environmental and Climate Research, Jinan University, Guangzhou 511443, China

[3]Guangdong-Hongkong-Macau Joint Laboratory of Collaborative Innovation for Environmental Quality, Jinan University, Guangzhou 510632, China

[4] State Key Laboratory of Advanced Environmental Technology, Guangzhou Institute of Geochemistry, Chinese Academy of Sciences, Guangzhou, 510640, China

[5] Guangdong-Hong Kong-Macao Joint Laboratory for Environmental Pollution and Control, Guangzhou Institute of Geochemistry, Chinese Academy of Science, Guangzhou, China

[6] Guangdong Provincial Academy of Environmental Science, Guangzhou, 510045, China

[7] Guangzhou Research Institute of Environment Protection Co.,Ltd, Guangzhou 510620, China

[8]School of Atmospheric Sciences, Guangdong Province Key Laboratory for Climate Change and Natural Disaster Studies, and Institute of Earth Climate and Environment System, Sun Yat-sen University, Zhuhai 519082, China

[9]Guangdong Environmental Monitoring Center, Guangzhou 510308, China
[†]Deceased, 10/2024

Correspondence to: Weiwei Hu (weiweihu@gig.ac.cn)

## **Abstract**

Secondary organic aerosol (SOA) is an important component of organic aerosol (OA), yet its atmospheric evolution and impacts on volatility remain poorly understood. In this study, we investigated the volatility of different types of SOA at a downwind site of the Pearl River Delta (PRD) region in the fall of 2019, using a time-of-flight chemical ionization mass spectrometer coupled with a Filter Inlet for Gases and Aerosol (FIGAERO-CIMS). Positive matrix factorization (PMF) analysis was performed on the thermogram data of organic compounds (referred as FIGAERO-OA) measured by the FIGAERO-CIMS. Eight factors were resolved, including six daytime chemistry related factors, a biomass burning related factor (BB-LVOA, 10% of the FIGAERO-OA), and a nighttime chemistry related factor (Night-LVOA, 15%) along with their corresponding volatility. Day-HNO$_x$-LVOA (12%) and Day-LNO$_x$-LVOA (11%) were mainly formed through gas-particle partitioning. Increasing NO$_x$ levels mainly affected SOA formation through gas-particle partitioning, suppressing the formation of low-volatile organic vapors, and thus promoting the formation of relatively high volatile OA with a higher N:C ratio. Two aged OA factors, Day-aged-LVOA (16%) and Day-aged-ELVOA (11%), were attributed to daytime photochemical aging of pre-existing OA. In addition, the daytime formation of Day-urban-LVOA (16%) and Day-urban-ELVOA (7%) could only observed in the urban plume. Results show that both gas-particle partitioning (36%) and photochemical aging (30%) accounted for a major fraction in FIGAERO-OA in the afternoon during the urban air masses period, especially for high-NO$_x$-like pathway (~21%). In general, the six daytime OA factors collectively explain the majority (82%) of daytime SOA identified by an aerosol mass spectrometer (AMS). While BB-LVOA and Night-LVOA accounted for 13% of biomass burning OA and 48% of nighttime chemistry OA observed by AMS, respectively. Our PMF analysis also demonstrated that the the highly oxygenated OA and hydrocarbon-like OA cannot be identified with FIGAERO-CIMS in this study. In summary, our results show that the volatility of OA is strongly governed by its formation pathways and subsequent atmospheric aging processes.

## 1. Introduction

Secondary organic aerosols (SOA), a major component of fine particular matter ($PM_{2.5}$) in China (Zhou et al., 2020), exert profound influences on climate change, human health, and air quality (Arias et al., 2021; Apte et al., 2018; Huang et al., 2014). Despite notable reductions in primary emission in recent years, SOA has emerged as an increasingly crucial factor in haze formation in China (Zhang et al., 2018). However, accurately modeling SOA from current chemical models is still challenging, largely attributed to our limited understanding of its formation mechanisms (Charan et al., 2019; Matsui et al., 2009; Lu et al., 2020). Thus, there is a crucial need for a comprehensive understanding of SOA formation and aging processes in the ambient environment.

Positive matrix factorization (PMF) has been widely used to apportion the contribution of primary and secondary sources to organic aerosol (OA) (Chen et al., 2014; Chen et al., 2021a; Ou et al., 2023; Tian et al., 2016). For the input of OA, the matrix of time serial spectral of OA measured by the Aerodyne Aerosol Mass Spectrometers (AMS) or Aerosol Chemical Speciation Monitor (ACSM) was usually applied (Uchida et al., 2019; Canonaco et al., 2013). Based on this approach, various primary OA (POA) components such as, hydrocarbon-like OA (HOA, associated with traffic emission), biomass burning OA (BBOA), cooking OA (COA), and secondary OA (SOA) with different oxidation levels are broadly identified in field measurements (Zhang et al., 2012; Jimenez et al., 2009; Huang et al., 2010; Qin et al., 2017; Guo et al., 2020; Huang et al., 2018; Al-Naiema et al., 2018). OA factors are generally distinguished according to their features on mass spectral and time series (Ulbrich et al., 2009; Lee et al., 2015). However, the electric ionization sources (70ev), together with thermal decomposition at 600C, lead to strongly fragmented ions detected in AMS/ACSM. These fragmented ions lack parent molecular information, thus hindering the ability to further attribute OA factors to more specific sources, thereby limiting our understanding of SOA formation pathways and aging mechanisms in ambient environments. To overcome this challenge, applying PMF analysis of molecular-level datasets is needed for refining SOA source apportionment. Recently, chemical ionization mass spectrometer coupled with the Filter Inlet for Gases and Aerosols (FIGAERO-CIMS) has been increasingly employed for the molecular-level characterization of oxygenated organics compounds in the gas and particle phase (Ye et al., 2021;

Thornton et al., 2020). Using this approach, Ye et al. (2023) employed PMF analysis to FIGAERO-CIMS data sets and found that low-NO-like pathway had a significant contribution to SOA formation in urban area.

Volatility, an important property of organic compounds, is frequently described as saturation mass concentration ($C^*$, Donahue et al., 2006). The volatility of organic compounds is closely related to its chemical characteristics, including oxidate state, number of carbons, and functional groups (Donahue et al., 2012; Donahue et al., 2011; Ren et al., 2022). The gas–particle partitioning behavior of organic compounds is largely governed by their volatility, and thus strongly influence the formation of SOA (Nie et al., 2022). Moreover, chemical processes occurring in the particle phase can alter the volatility of organic compounds. For example, high molecular weight organic compounds can form through accretion reactions, leading to a reduction in volatility (Barsanti and Pankow, 2004; Jenkin, 2004; Kroll and Seinfeld, 2008). In addition, particle phase organic compounds can be oxidized by atmospheric oxidants (e.g., $O_3$, OH, and $NO_3$), which can also alter the chemical characteristic and volatility (Rudich et al., 2007; Walser et al., 2007). Thus, the variation of volatility can provide valuable information about the formation and aging processes of OA. Graham et al. (2023) found that SOA from $NO_3$ oxidation of α-pinene or isoprene had a higher volatility than it from β-caryophyllene. Hildebrandt Ruiz et al. (2015) demonstrated that exposure to different OH levels could lead to a large variation in SOA volatility.

However, linking OA volatility directly to its chemical characteristics and sources remains challenging. A thermodenuder (TD) coupled with an AMS has been employed to investigate the volatility of OA from different sources (Louvaris et al., 2017). Xu et al. (2021) estimated the volatility of different PMF OA factors in the North China Plain and reported that RH level could alter both the formation pathway and volatility of more oxidized oxygenated OA. Feng et al. (2023) reported the much lower OA volatility from out plumes of North China plain than results obtained in the urban areas, signifying the aging impact on OA volatility. Nevertheless, owing to the operational principle of AMS, it is still difficult to obtain molecular information of organic compounds at different volatilities. In contrast, the FIGAERO-CIMS provides not only molecular-level measurements but also thermal desorption profiles (thermograms) for each detected compound. The temperature of the peak desorption signal ($T_{max}$) of a specific compound is typically correlated with its volatility (Lopez-Hilfiker et al., 2014), enabling direct connects between the volatility and

organic molecular (Ren et al., 2022). Huang et al. (2019) analysis the ambient particles filter samples
collected in different seasons with FIGAERO-CIMS and reported a lower volatility of oxygenated
OA in winter, partly due to higher O:C. Buchholz et al. (2020) utilized PMF analysis of FIGAERO-
CIMS thermogram data sets to investigate physicochemical property of laboratory-generated SOA
particles.
To comprehensively investigate the evolution of OA and its relationship with volatility in
ambient environment, we employed a FIGAERO-CIMS along with other online instruments to gain
a comprehensive understanding of the variation in SOA volatility within urban plumes in the Pearl
River Delta (PRD) region during the fall of 2019. PMF analysis was performed on thermograms
data obtained from the FIGAERO-CIMS. By combining the source apportionment of thermogram
organic aerosol (OA) with corresponding volatility information, we investigated the potential
formation pathway and influencing factors of SOA in the urban downwind region.

## 125 2.Measurement and Method

### 126 2.1 Field Measurements

We conducted a field campaign at the Heshan supersite in the PRD region from September 29
to November 17, 2019. Considering the integrity of the measurements, we focus primarily on the
period from October 16 to November 16, 2019 in this study. The measurement site was located in a
rural area surrounded by farms and villages (at 22°42′39. 1″N, 112°55′35.9″E, with an altitude of
about 40 m), situated to the southwest of the PRD region. All online instruments were placed in air-
conditioned rooms on the top floor of the supersite building.
A FIGAERO-CIMS, coupled with an X-ray source, was used to measure organic compounds
in both the gas- and particle-phase, utilizing I⁻ as the chemical ionization reagent. The instrument
operated on one-hour cycle by switching between two modes (sampling mode and desorption mode)
for measuring gas- and particle-phase oxygenated organic molecules. In the sampling mode,
ambient gas was measured in the first 21 minutes, followed by a 3-min zero air background, while
the $PM_{2.5}$ sample was collected on a PTFE membrane filter for 24 minutes. Then, the instrument
was switched to the desorption mode, in which the collected particles were desorbed using heated
$N_2$. The temperature of the $N_2$ was increased from approximately 25°C to 175°C over a 12-minute
period, and then held at 175°C for an additional 24 minutes. Calibration of a few chemicals was
conducted in the laboratory. For the remaining organic species, a voltage scanning method was used
to determine their sensitivities (referred to as semi-quantified species) (Ye et al., 2021; Iyer et al.,
2016; Lopez-Hilfiker et al., 2016). The detailed operation settings, data processing, and calibration
can be found in Cai et al. (2023) and Ye et al. (2021).
A soot particle aerosol mass spectrometer (SP-AMS, Aerodyne Research, Inc., USA) was used
to measure the chemical composition of $PM_1$ particles, including nitrate, sulfate, ammonium,
chloride, black carbon, and OA. More details on the quantification using ionization efficiency,
composition dependent collection efficiency, data analysis,  and source apportionment of OA from
AMS data (defined as AMS-OA) could be found in Kuang et al. (2021) and Cai et al. (2024). In
brief, AMS-OA consisted of two primary OA factors and four secondary OA factors. The primary
OA factors include hydrocarbon-like OA (HOA, 11%) and biomass burning OA (BBOA, 20%),
which were mainly contributed by traffic and cooking emissions and biomass burning combustion,
respectively. For SOA factors, biomass burning SOA (BBSOA, 17%) was likely formed through
oxidation of biomass burning emission; less oxidized oxygenated OA (LO-OOA, 24%), which
results from strong daytime photochemical processes; more oxidized oxygenated OA (MO-OOA,
17%), related to regional transport; and nighttime-formed OA (Night-OA, 11%) which was
associated with nighttime chemistry.
Trace gases such as $O_3$ and $NO_x$ were measured by gas analyzers (model 49i and 42i, Thermo
Scientific, US). Meteorological parameters, including wind speed and wind direction, were
measured by a weather station (Vantage Pro 2, Davis Instruments Co., US).
**2.2  Methodology**
Positive matrix factorization (PMF) is a widely used tool for source apportionment of long
timeseries data (Paatero and Tapper, 1994). In the desorption mode, the particulate organic
compounds are thermo-desorbed and simultaneously measured by the FIGAERO-CIMS. Organic
molecules with different volatility were characterized by thermograms (desorption signals vs
temperature of $N_2$). Here, we performed PMF analysis to the thermogram data of organic
compounds measured by the FIGAERO-CIMS (FIGAERO-OA) using the Igor-based PMF
Evaluation Tool (PET, v3.01, Ulbrich et al., 2009), which can be expressed as follows:
$$X = GF + E \tag{1}$$
where $X$ is the thermogram organic compound data measured by the FIGAERO-CIMS. which

can be decomposed into two matrices $G$ and $F$. The matrix $G$, $F$, and $E$ contain the factor time series, factor mass spectra, and the residuals between the measured data and the reconstructed data.

The raw normalized count per second (NCPS) thermogram data with a time resolution of 1s was averaged to a 20s-time grid, and then was background-corrected by subtracting linearly interpolated background thermogram signals. For each scan, only the data points when the desorption temperature increased were used as input data (corresponding to 25°C to 170 °C,1-70 data points in this study, Fig. S1), since the main information lies during the species desorbing from the FIGAERO filter (Fig. S1, Buchholz et al., 2020). Then, we combined data from separate thermogram scans (without background scans) to a larger input data set.

To perform the PMF analysis, a data uncertainty matrix ($S_{i,j}$) is needed, where the $i$ and $j$ represents the index of ions and data points, respectively. According to Buchholz et al. (2020), the uncertainty was assumed to be constant for each individual thermogram scan (constant error scheme). The $S_{i,j}$ of a specific thermogram scan can be determined by the following equation:

$$S_{i,j} = \sigma \tag{2}$$

For each thermogram scan, the last 20 data points are assumed to be in steady state. Thus, the $\sigma_{noise}$ was calculated as the median of the standard deviation of the residual ($res_{i,j}$), which can be obtained from the difference between the data points ($Data_{i,j}$) and the corresponding linear fitted value ($FittedData_{i,j}$) for the measured data points (Fig. S2):

$$res_{i,j} = Data_{i,j} - FittedData_{i,j} \tag{3}$$

$$\sigma = median(stdev(res_{i,j})) \tag{4}$$

Due to the large volume of the data matrix (59500×1028) exceeding the processing capacity of the PET, we had to divide the data matrix into three parts and performed PMF analysis separately. An eight-factor solution was selected for each part based on $Q/Q_{exp}$ behavior and factor interpretability (Fig. S3 to S6). To assess factor consistency, the mass spectra of resolved factors were compared across different parts, showing strong correlations (R>0.9) for the each factor (Fig. S7 and S8). Weaker correlations during the early campaign period (2 to 5 October) likely reflect changes in dominant OA sources under different meteorological conditions (Fig. S8 and S9). After excluding this period, consistent factor profiles were obtained and combined for further analysis. Detailed evaluations are provided in the Section S1.

Since input data sets of PMF analysis were the NCPS data, the signal of each thermograms

factor was a combination of NCPS values of different ions. Thus, it is necessary to convert the signal of these factors into mass concentrations, which would increase the representativeness of the thermogram PMF results. The NCPS of a specific ion was linearly correlated with the corresponding mass concentration. Thus, for a signal running cycle (a thermogram scan), the mass concentration of a specific thermograms OA factor $k$ ($M_k$) can be estimated as:

$$M_k = \sum_i \left( \frac{\sum_j Signal_{j,k} \cdot Profile_{i,k}}{\sum_j NCPS_{j,i}} \cdot m_i \right) \tag{5}$$

where $i$ and $j$ represent the index of species and data points; the $Signal_{j,k}$ is the signal of a thermograms OA factor $k$ at a data index $j$; the $Profile_{i,k}$ represents the fraction of signal of factor $k$ and ion $i$; the $NCPS_{j,i}$ is the NCPS of species $i$ at a data index $j$; and $m_i$ is the mass concentration of species $i$ in the particle-phase measured by the FIGAERO-CIMS.

For a specific organic compound, the temperature of the peak desorption signal ($T_{max}$) has a nearly linear relationship with the logarithm of saturation vapor pressure ($P_{sat}$) of the respective organic compound (Lopez-Hilfiker et al., 2014):

$$ln(P_{sat}) = aT_{max} + b \tag{6}$$

where $a$ and $b$ are fitting coefficients. $P_{sat}$ can be converted to saturation vapor concentration ($C^*$, μg m⁻³) by following equation:

$$C^* = \frac{P_{sat}M_w}{RT}10^6 \tag{7}$$

where $M_w$ is the average molecular weight of the organic compound (determined by the FIGAERO-CIMS), R is the gas constant (8.314 J mol⁻¹ K⁻¹), and T is the thermodynamic temperature (298.15 K). The fitting parameters of $a$ and $b$ were calibrated by a series of polyethylene glycol (PEG 5-8) compounds before the campaign. PEG standards (dissolved in acetonitrile) were atomized using a homemade atomizer, and the resulting particles were size-classified by a differential mobility analyzer (DMA; model 3081L, TSI Inc.) to target diameters of 100 and 200 nm. The size-selected particles were then split into two flows: one directed go to a CPC (3775, TSI) for the measurements of number concentration, and the other to the FIGAERO-CIMS particle inlet. The collected mass by CIMS was calculated based on the particle diameter, number concentration, FIGAERO-CIMS inlet flow rate, and collection time. The details of the calibration experiments and selection of fitting coefficients (a and b) can be found in table S1 and Cai et al. (2024). In this study, the fitting parameters (a=-0.206 and b=3.732) were chosen, as the mass loading (407 ng) and diameter (200

nm) are closest to the ambient samples, since the collected mass loading centered at about 620 ng and the particle volume size distribution (PVSD) centered at about 400 nm (Cai et al., 2024). It was worth noting that the volatility range of PEG 5-8 ($-1.73 \leq \log_{10} C^* \leq 3.34$ μg m$^{-3}$) may not fully represent the volatility of ambient organic aerosol, particularly nitrogen-containing and highly oxygenated compounds that can exhibit much lower volatility ($\log_{10} C^* \leq$-2 μg m$^{-3}$, Ren et al., 2022; Chen et al., 2024). At present, saturation vapor pressure data for PEG standards are only available up to PEG-8 (Krieger et al., 2018). Ylisirniö et al. (2021) demonstrated that different extrapolation approaches for estimating the volatility of higher-order PEGs can lead to substantial discrepancies in calibration results, and they strongly recommended that higher-order PEGs should only be used to extend the volatility calibration range once their saturation vapor pressures are accurately determined. Very recently, Ylisirniö et al. (2025) derived saturation vapor pressures for higher-order PEGs up to PEG-15 and demonstrated that extending FIGAERO-CIMS calibration to much lower volatilities is feasible, but also showed that different estimation approaches for higher-order PEGs can lead to large discrepancies, highlighting substantial uncertainties when extrapolating volatility calibration beyond PEG-8. Therefore, uncertainties may remain in the calibration of low-volatility OA, and further calibration experiments using complementary techniques are highly recommended. Therefore, uncertainties may remain in the calibration of low-volatility OA, and further calibration experiments using complementary techniques are highly recommended.

## 3 Results

### 3.1 Overview of FIGAERO-OA factors

In this study, the average mass concentration of FIGAERO-OA was about 5.3±2.4 μg m$^{-3}$. The thermogram data sets of FIGAERO-OA were analyzed with PMF and mass concentration of each factor was estimated based on eq. (5), which provide volatility and mass concentration information of OA originating from different formation pathways. An 8-factor solution was chosen to explain the thermogram of FIGAERO-OA. These factors included six associated with daytime photochemical reactions, one related to biomass-burning, and one factor contributed by nighttime chemistry. The diurnal variation, mass spectra, and thermograms of these factors can be found in Fig. S9. The estimated volatility ($\log_{10} C^*$), $T_{max}$, and elemental information of all factors are shown in table 1. Given that the thermogram data can provide volatility information of organic

compounds, the identified OA factors were categorized based on their potential formation pathway,
volatility, and correlation with AMS PMF factors (Table 1 and Fig. S10). For example, if the PMF
factor with a $T_{max}$ located in the ranges of low volatile organic compounds (LVOC, approximately
corresponding to 78.8 ℃ to 112.3 ℃ then Fig. 1), this factor will be named after low volatility OA
(LVOA). For the factors whose $T_{max}$ is above 112.3 ℃, extremely low volatility (ELVOA) will be
named.

The six daytime chemistry related factors include a low volatility OA factor likely formed

under high $NO_x$ condition (Day-HNO$_x$-LVOA, 12%), a low volatility factor contributed by gas-
particle partitioning (Day-LNO$_x$-LVOA, , 11%), a low volatility and an extremely low volatility
factor originating from the daytime aging process (Day-aged-LVOA and Day-aged-ELVOA, 16%
and 11% respectively), and a low volatility and an extremely low volatility factors related to urban
air masses (Day-urban-LVOA and Day-urban-ELVOA, 16% and 7%, respectively). These daytime
factors accounted for about 76.4% of the total mass of FIGAERO-OA and demonstrated distinct
daytime peak. The total mass of daytime FIGAERO-OA factors showed a strong positive correlation
with LO-OOA in AMS-OA (R=0.86), which was attributed to photochemical reactions (Fig. S10a).

Both Day-HNO$_x$-LVOA and Day-LNO$_x$-LVOA reached their peak values at about 14:00 LT

(Fig. 1 a1 and b1), implying strong photochemical production. Day-HNO$_x$-LVOA had the highest
N:C (0.06) and the lowest oxidation state ($\overline{OS_c}$=-0.01), which could be attributed to the "high $NO_x$"
formation pathway. It was also supported by significant positive correlation (R=0.93-0.94) with
particulate phase nitrogen-containing organic compounds (e.g., $C_4H_5NO_6$, $C_8H_{11}NO_8$, and
$C_8H_{11}NO_9$). Previous studies found that high $NO_x$ concentration can suppress the production of
molecules with a high oxidation degree (Rissanen, 2018; Praske et al., 2018), which could explain
the low $\overline{OS_c}$ value (-0.01) and relative high volatility ($\log_{10} \overline{C^*}$=-0.98) found for Day-HNOx-LVOA.
Day-LNO$_x$-LVOA had a higher $\overline{OS_c}$ (0.18) and lower $\log_{10} \overline{C^*}$ (-2.71) than Day-HNO$_x$-LVOA,
consistent with that Day-LNO$_x$-LVOA was composed of smaller and more oxidized non-nitrogen
containing compounds (e.g., $C_2H_2O_3$, $C_3H_4O_3$, $C_4H_6O_4$, and $C_6H_8O_4$). Noting that C2-C3 group
could originate from the decomposition of larger molecules during thermal desorption, since the
thermogram of $C_2H_2O_3$ and $C_3H_4O_3$ demonstrated a bimodal distribution (Fig. 9 a). Figure S12 b
and d further examine the contribution of all FIGAERO factors to the signals of $C_2H_2O_3$ and $C_3H_4O_3$.
One mode, peaking in the LVOC range, was primarily associated with Day-LNO$_x$-LVOA, and a
second mode, peaking in the ELVOC range, was dominated by Day-aged-ELVOA. These results
indicates that these two low molecular weight species are likely decomposition products of at least
two distinct classes of higher molecular weight organic compounds.
Additionally, we identified two aged OA factors (Day-aged-LVOA and Day-aged-ELVOA)
with an afternoon peak at about 18:00 LT (Fig. 1 c1 and d1), which may be derived from the aging
transformation of preexisting organic aerosols via daytime photochemical reactions. These aged
factors exhibited the highest $\overline{OS_c}$ (0.35 and 0.40) and relatively low volatility with a $\log_{10}\overline{C^*}$ of -
2.02 and -4.80, respectively. Day-aged-LVOA was featured with a series of $C_4$-$C_8$ oxygenated
compounds (e.g., $C_4H_6O_5$, $C_5H_8O_5$, $C_6H_{10}O_5$, $C_7H_{10}O_5$, and $C_8H_{12}O_5$). In contrast, Day-aged-
ELVOA had a higher fraction of smaller molecules (e.g., $C_2H_4O_3$ and $C_3H_6O_3$, Fig. 1d2). Chen et
al. (2021b) found that low molecular weight carboxylic acids (LMWCA) could form through SOA
aging processes and report a strong correlation ($R^2$=0.90) between LMWCA and highly oxygenated
OA. However, $C_2H_4O_3$ and $C_3H_6O_3$ had a weak correlation (R=0.49 and 0.13) with MO-OOA
resolved from AMS (Fig. S11). In addition, the $T_{max}$ of $C_2H_4O_3$ and $C_3H_6O_3$ located in the ELVOC
range and showed thermogram profiles similar to that of Day-aged-ELVOA (Fig. S12a). The
thermogram signal of $C_2H_4O_3$ and $C_3H_6O_3$ was mainly contributed by Day-aged-ELVOA (Fig. S12
c and e), supporting the interpretation that these species are more likely decomposition products of
low volatility organic compounds rather than being directly formed through atmospheric aging
processes.
Two urban air masses-related OA factors (Day-urban-LVOA and Day-urban-ELVOA) were
identified, which would be discussed in the following section. Day-urban-LVOA demonstrated
comparable $\overline{OS_c}$ (0.08), O:C (0.80) and volatility (-0.90) to Day-HNO$_x$-LVOA (-0.01, 0.75, and -
0.98, respectively), but show a higher fraction of non-N-containing molecules (e.g., $C_4H_6O_4$,
$C_5H_6O_4$, $C_5H_8O_5$, and $C_7H_{10}O_5$) and a reduced N:C ratio (Table 1). However, the oxidation state
($\overline{OS_c}$) of Day-HNO$_x$-LVOA (-0.01) was significantly lower than that of Urban-LVOA (0.08) ,
accompanied by a relatively higher N:C (0.06 vs 0.04). Despite its lower oxidation state, the
volatility of Day-HNO$_x$-LVOA is comparable to that of Day-urban-LVOA, which may reflect
differences in functional group composition. For example, a nitrate group (-ONO$_2$) contributes to
volatility reduction at a level comparable to that of a hydroxyl group (-OH) and generally more
strongly than carbonyl functionalities such as aldehydes (–C(O)H) or ketones (–C(O)–) (Pankow
and Asher, 2008). However, due to instrumental limitations, we are unable to directly resolve the
functional group composition of individual OA factors, and further measurements employing new
techniques are needed to better constrain the role of functional groups in controlling the volatility
of ambient organic aerosol. Day-urban-ELVOA had the lowest volatility ($\log_{10} C^* = -7.18$) but an
$\overline{OS_c}$ (0.27) lower than Day-aged-ELVOA (0.34) and composed of oxygenated compounds (e.g.,
$C_8H_{10}O_5$, $C_7H_8O_5$, $C_6H_8O_4$, and $C_5H_6O_4$). The thermogram of Day-aged-ELVOA demonstrates
bimodal distribution (peaked at LVOC and ELVOC range) and had a highest $T_{max}$ (153.2 °C)
among thermograms OA factors (Fig. 1f3). However, the majority of organic molecules (e.g.,
$C_5H_6O_4$, $C_4H_6O_5$, $C_6H_8O_4$, and $C_8H_{10}O_5$) do not exhibit thermograms similar to that of Day-urban-
ELVOA (Fig. S13). Instead, their thermograms demonstrate multimodal distributions and are
contributed by multiple FIGAERO factors. For example, a mode of $C_5H_6O_4$ peaking in the LVOC
range was mainly contributed by Day-urban-LVOA, while two modes peaking in the ELVOC range
were primarily contributed by Day-aged-ELVOA and Day-urban-ELVOA, respectively. These
results suggest that these molecules may originate from both direct desorption of organic aerosol
and thermal decomposition of higher-molecular-weight compounds during heating.
The biomass-burning related factor, biomass-burning less volatile organic aerosol (BB-LVOA,
10% of FIGAERO-OA), had a low $\overline{OS_c}$ (-0.07), the lowest O:C (0.74), and positive correlation with
BBOA resolved from AMS (R=0.64, Fig. S10b). It presented a prominent peak at 19:00 LT and was
identified by the distinctive tracer levoglucosan ($C_6H_{10}O_5$), nitrocatechol ($C_6H_5NO_4$), and
nitrophenol ($C_6H_5NO_3$, Fig. 1) in the spectrum, which was frequently detected in biomass burning
plumes (Gaston et al., 2016; Ye et al., 2021). In the upwind urban region of Heshan site, Ye et al.
(2023) identified a biomass burning related factor in Guangzhou city using the FIGAERO-CIMS,
with a distinct evening peak at 21:00 LT and more abundance oxygenated compounds (e.g., $C_7H_{10}O_5$
and $C_8H_{12}O_6$). The different oxidation level of BBOA between Guangzhou and Heshan, suggest the
BB-LVOA in this study is more related to the direct BB emission, but the BB factor in Guangzhou
is more resembled BB related SOA factor. This statement was supported by the fact that biomass
burning activities were frequently observed near the measurement site during this study, while the
biomass burning activities in urban areas was prohibited and can be transported from nearby
suburban agricultural areas (Cai et al. 2023) .

The nighttime chemistry related less volatile OA (Night-OA, 15% of FIGAERO-OA) has the

highest N:C (0.07) and exhibited an enhanced at nighttime (22:00-24:00 LT, Fig. 1). Notably, this
nighttime factor was composed of a series of organic nitrates (e.g., $C_8H_{11}NO_7$, and $C_{10}H_{15}NO_7$),
which was related to the products from monoterpenes oxidized by the $NO_3$ radical or oxidation of
biomass burning products during nighttime (Faxon et al., 2018; Decker et al., 2019). Noting that the
levoglucosan ($C_6H_{10}O_5$) was also abundant in the Night-LVOA, suggesting that part of this factor
could be attributed to the nighttime aging process of biomass burning products (Jorga et al., 2021).
The detailed discussion about the potential formation pathway of these six daytime FIGAERO-OA
factors will be discussed in section 3.2.

The volatility of organic compounds was closely related to their chemical characteristics

(Donahue et al., 2012). Figure 2 demonstrates the relationship between $\log_{10} \overline{C^*}$ of thermogram
factors and $\overline{OS_c}$, O:C, and number of carbons ($nC$).  In general, these factors exhibited a negative
correlation (R=-0.60 and -0.73) with both the $\overline{OS_c}$ and the O:C but showed a positive correlation
(R=0.73) with $nC$ (Fig. 2), except for Day-urban-ELVOA. As aforementioned, the major component
of Day-urban-ELVOA could be decomposition products of larger oxygenated molecules. Thus, the
chemical characteristic of Day-urban-ELVOA did not demonstrate a similar relationship of volatility
versus molecule indicators (e.g., oxidation state, O:C and $nC$)  as other factors. The increase of
carbon number usually lead to a decrease in volatility (Donahue et al., 2011), while this trend
overturned in this campaign (Fig. 2c). Fig. 2d shows that $\overline{OS_c}$ had a negative relationship (R=-0.84)
with carbon number, suggesting that organic factors with a higher oxidation degree had a shorted
carbon backbone. It could be partly owing to fragmentation of organic molecules during aging
processes (Chacon-Madrid and Donahue, 2011; Jimenez et al., 2009). Consistently, two aged factors
(Day-aged-LVOA and Day-aged-ELVOA) had a higher $\overline{OS_c}$ and a lower carbon number than other
factors. Additionally, it indicates that the increase in oxidation degree outweighed the effect of
decreasing $nC$, leading to a reduction in the volatility of OA during this campaign.

The temporal variation of volatility distribution and mean $C^*$ of FIGAERO-OA, the sum of

six daytime factors in FIGAERO-OA and LOOA in AMS OA, mass fraction of eight FIGAERO-
OA factors, and wind direction and speed are demonstrated in Fig. 3. As shown in Fig. 3, the
increase of mean $C^*$ during the daytime (6:00 LT to 18:00 LT, Fig. S14 a) is usually accompanied
by the enhancement of daytime factors in FIGAERO-OA and LO-OOA from AMS (Fig. S14 b and
c), indicating that the formation of these factors could notably increase OA volatility. Notably,
FIGAERO-OA with a $\log_{10} C^*$ of -1 μg m$^{-3}$ showed pronounced enhancements during the
increasing of mean $C^*$, implying that the volatility of six daytime factors might cluster around $10^{-1}$
μg m$^{-3}$ (Fig. S14d). In Fig. 3b, distinct diurnal variation of O$_x$ (O$_x$=O$_3$+NO$_2$) was observed during
the campaign. The maximum of Ox can be as high as 230 ug m$^{-3}$, highlighting strong photochemical
reaction. The daytime factors, especially Day-HNO$_x$-LVOA (Fig. 3c), exhibited markable
enhancements under weak northwesterly to northeasterly wind (Fig. 3d and Fig. S15). A backward
trajectory analysis revealed that the measurement site was mainly affected by the urban pollutants
from the city cluster around Guangzhou (Fig. S16). Two periods, which were long-range transport
and urban air massed periods, respectively, were selected to further analyze the impact of urban
pollutants on the formation and aging process of OA. The variation of OA volatility based on wind
direction andspeed, together with backward trajectory analysis, were also explored (Fig. S16 and
Table S2). In general, during the urban air masses period, the site was influenced by regional urban
plumes from the northeast city cluster, while the long-range transport period was primarily
associated with air masses advected from the northeast inland regions. More detailed discussion will
be shown in the following section.
**3.2 Potential formation pathway of FIGAERO-OA**
Figure 4 demonstrates distinct differences in the diurnal variation of thermograms factors
(including Day-HNO$_x$-LVOA, Day-aged-LVOA, Day-urban-LVOA, and Day-urban-ELVOA)
during long-range transport period and urban air masses period. During the urban air masses period,
Day-HNO$_x$-LVOA significantly increased from ~0.4 μg m$^{-3}$ to ~4.8 μg m$^{-3}$ in the daytime. The mass
concentration of Day-urban-LVOA and Day-urban-ELVOA demonstrated daytime enhancements
only during urban period, suggesting that the formation of these factors was closely related to the
pollutants in the urban plumes. Consistently, During the urban air masses period, the maximum
ozone concentration in the afternoon (12:00-18:00 LT, 208.3 μg m$^{-3}$) was higher than that (185.5
μg m$^{-3}$) during long-range transport period, indicating a stronger photochemical reaction in the urban
plumes (Fig. 4). Thus, the daytime thermogram factors accounted for a higher fraction (79% vs 75%)
of FIGAERO-OA (Fig. S17). Additionally, the average mass concentration of all thermogram
factors (8.9±5.1 μg m$^{-3}$) was noticeably increased compared to the long-range period (5.3±2.4 μg
m$^{-3}$). Elevated NO$_x$ concentration was observed in the urban plumes in the afternoon (12:00 LT-
18:00 LT, 17.4 ppbv vs 11.7 ppbv), which might also affect the formation pathway of SOA. Both
NO and NO/NO$_2$ remained at a relative low level (0.6-0.8 ppbv and <0.5) in the afternoon during
these two periods (Fig. S18), suggesting an important role of low-NO-like pathway (Ye et al., 2023).
Nihill et al. (2021) found that the production of OH and oxidized organic molecules would be
suppressed under high NO/NO$_2$ (>1) condition. Notably, Day-HNO$_x$-LVOA accounted for the
largest portion (29%) of FIGAERO-OA in the afternoon (12:00-18:00 LT, Fig. S19), followed by
Day-aged-LVOA (21%), while Day-LNO$_x$-LVOA contributed only 6%. In contrast, during the
long-range transport period, the mass fraction of Day-LNO$_x$-LVOA significantly increased (from
6% to 15%) along with a decrease in Day-HNO$_x$-LVOA (from 29% to 21%). These results indicate
that elevated NO$_x$ concentration in urban plumes might alter the formation pathway of SOA (Cai et
al., 2024). Note that the sum of six daytime FIGAERO factors showed a positive relationship
(R=0.80 and 0.76, respectively) with LOOA during both periods (Fig. S20). However, the slope
(0.81) of the linear regression during the urban air masses period was higher than that (0.58) during
the long-range transport period, indicating that a higher fraction of LOOA could be detected by the
FIGAERO-CIMS during urban air masses period. This difference could be related to the
discrepancy in OA volatility. According to Cai et al. (2024), the volatility of OA was higher during
the urban air masses period.

To explore the potential formation pathway of daytime factors, figure 5 demonstrates the

variation of mass concentrations of six daytime factors as a function of O$_x$, total gas-phase organic
molecules measured by the FIGAERO-CIMS (referred as organic vapors), and NO$_3^-$/SIA. Five
factors, excluding Day-urban-LVOA, exhibited positive correlations with O$_x$, highlighting the
critical role of photochemical reactions in their formation. Previous studies have demonstrated that
gas–particle partitioning plays a key role in SOA formation (Nie et al., 2022; Wang et al., 2022). In
this study, organic vapors had strong positive correlations with Day-HNO$_x$-LVOA (R=0.73) and
Day-LNO$_x$-LVOA (R=0.74), suggesting that these factors were mainly formed via gas-particle
partitioning. The median concentration of Day-HNO$_x$-LVOA dramatically increased (from ~0 to
~5.6 μg m$^{-3}$) with rising organic vapors, whereas a comparable enhancement was not observed for
Day-LNO$_x$-LVOA (Fig. 5 b1 and b2).

Furthermore, NO$_x$ impact on Day-HNO$_x$-LVOA and Day-HNO$_x$-LVOA was investigated here.

Fig. S21 show Day-HNO$_x$-LVOA concentrations were consistently higher under elevated NOx

conditions, while Day-HNO$_x$-LVOA decreased with increasing NO$_x$ level.  Figure 6a displays the mass ratio of Day-HNO$_x$-LVOA to Day-LNO$_x$-LVOA obviously increased with organic vapors (up to 12~26) under high NO$_x$ condition (>20 ppbv), while the ratio remained at approximately 2 at low NO$_x$ level (<10 ppbv). These overall results suggest that Day-HNO$_x$-LVOA formation was predominantly governed by gas-particle partitioning under high NO$_x$ condition, which were typically sustained during urban air masses period (Fig. 2d). Figure 6b compares the relative mass fraction of molecular composition in two gas-particle partitioning related factors, Day-HNO$_x$-LVOA and Day-LNO$_x$-LVOA. The mass fraction of species was derived from the signal profile of corresponding factors based on their sensitivity (Ye et al., 2021). Day-HNO$_x$-LVOA presented greater proportions ($10^{-5}$~$10^{-3}$) of organic nitrates (ONs) than Day-LNO$_x$-LVOA ($10^{-11}$~$10^{-9}$), including C$_4$H$_7$NO$_6$, C$_8$H$_9$NO$_4$, C$_8$H$_{11}$NO$_7$, as well as nitrophenols (e.g., C$_7$H$_7$NO$_3$), which are characterized by relatively low $\overline{OS_C}$. These compounds were probably attributed to the SOA formation under elevated NO$_x$ concentration (Fig. 2d). In contrast, Day-LNO$_x$-LVOA was enriched in non-nitrogen-containing species (e.g., C$_4$H$_6$O$_3$, C$_5$H$_{10}$O$_3$, C$_{11}$H$_{17}$O$_6$), which exhibited a higher $\overline{OS_C}$. These results indicate that NO$_x$ exerts contrasting effects on the formation of these two gas–particle partitioning-related factors.

Previous studies show that NO$_x$ has a nonlinear effect on the formation of highly oxygenated organic (HOM) compounds by influencing the atmospheric oxidation capacity and RO$_2$ autoxidation (Xu et al., 2025; Pye et al., 2019; Shrivastava et al., 2019). NO$_x$ could suppress the production of low-volatility molecules by inhibiting autoxidation (Rissanen, 2018; Praske et al., 2018), while Nie et al. (2023) found that NO could enhance the formation of HOM at low NO condition (< 82 pptv). During this campaign, the average NO$_x$ and NO was about 24.0 ppbv and 2.3 ppbv, respectively, substantially higher than the "low-NO-regime" described by Nie et al. (2023). Our previous study reported a lower concentration of organic vapors with a high $\overline{OS_c}$ within urban plumes during the same campaign (Cai et al., 2024). We investigate diurnal evolution of organic compositions under long-range transport and urban air masses periods (Fig. S22). Mass concentrations of CHON increase during the daytime in both periods, with a more pronounced enhancement observed in urban air masses (Fig. S22a). However, the mass fraction of CHON was lower during the urban air masses period than during the long-range transport period. We speculated that elevated NO$_x$ enhances overall oxidation and product formation rather than selectively

enriching nitrogen-containing compounds. This interpretation is consistent with results from our
previous observation-constrained box-model simulations, in which production rates of OH and
organic peroxyl radicals ($RO_2$) were evaluated under varying NOx and VOC conditions (Cai et al.,
2024). The modeled $P$(OH) were close to the transition regime, indicating that elevated $NO_x$ can
enhance atmospheric oxidation capacity. In contrast, the $P(RO_2)$ was in the VOC-limited regime
and decreased with increasing $NO_x$. Consistent with these results, Fig. S22c shows that the mass
fraction of highly oxygenated organic molecules (O$\geqslant$6) is lower during urban air masses period.
Concurrently, species with low oxygen numbers (O$\leqslant$3) become relatively more abundant in the
urban plumes (Fig. S22c), indicating a shift in the oxidation product distribution toward less
oxygenated and potentially more volatile compounds, the $NO_x$-driven suppression of
multigenerational autoxidation inferred from the box-model results. This suppression of oxidation
is observed for both CHON and CHO species. The average O:C of CHON (Fig. S22b) and CHO
(Fig. S22e) are both lower during the urban air masses period, suggesting that enhanced $NO_x$ broadly
suppresses autoxidation across organic compound classes.
Furthermore, as illustrated in Fig. S23, the mass concentration of SVOC (-0.5 <$\log_{10} C^*$< 2.5
µg m$^{-3}$) and LVOC (-3.5 <$\log_{10} C^*$< -0.5 µg m$^{-3}$, Donahue et al., 2012) in the gas phase exhibited
an increase (2.5 µg m$^{-3}$ at $NO_x$< 10 ppbv vs 3.3 µg m$^{-3}$ at $NO_x$$\geq$ 30 ppbv) with the increase in $NO_x$,
suggesting that these species likely contributed to the formation of Day-HNO$_x$-LVOA. Xu et al.
(2014) found that both SOA volatility and oxidation state exhibited a nonlinear response to $NO_x$ in
a series of chamber environment. SOA volatility decreases with increasing $NO_x$ level when the ratio
of initial NO to isoprene was lower than 3. At higher $NO_x$ level, higher volatile SOA was produced,
probably owing to the more competitive $RO_2$+NO pathway. Figure 5 c1 and c2 investigate the
relationship between these two factors and NO$_3^-$/SIA. Day-HNO$_x$-LVOA had a weak correlation
(R=0.30) with NO$_3^-$/SIA, while this trend overturned (R=-0.35) for Day-LNO$_x$-LVOA. Yang et al.
(2022) showed that OH+NO$_2$ pathway mainly contribute to the formation of nitrate in this campaign.
Together, these results indicate that elevated $NO_x$ suppressed the formation of highly oxygenated
organic compounds, thereby limiting the contribution to Day-LNO$_x$-LVOA. Thus, the Day-LNO$_x$-
LVOA was more likely formed via gas-particle partitioning under relatively low $NO_x$ condition.
It is worth noting that $C_4H_7NO_5$, likely originating from isoprene photooxidation in the
presence of $NO_x$ (Fisher et al., 2016; Paulot et al., 2009), also show a higher fraction in Day-LNO$_x$-
LVOA ($9.36 \times 10^{-5}$ vs $4.93 \times 10^{-11}$ in Day-HNOx-LVOA). A plausible explanation is that Heshan site,
located at a suburban region, experienced ambient $NO_x$ levels (~13 ppb in the afternoon) sufficient
to facilitate the formation of $C_4H_7NO_5$. It is further supported by the observation that both particle-
and gas-phase $C_4H_7NO_5$ showed no significant variation with increasing $NO_x$ (Fig. S24).
For the two urban-related factors, a positive correlation with $O_x$ was observed only during the
urban air masses period (R=0.46 and 0.64 vs -0.05 and 0.28 in the long-range transport period, Fig.
20 a and c). Notably, Day-urban-LVOA increased from ~1.0 to ~2.6 $\mu g\ m^{-3}$ as $O_x$ rose from 75 to
275 $\mu g\ m^{-3}$ during this period, while it remained relatively stable (~0.4 $\mu g\ m^{-3}$) during the long-
range transport period (Fig. S25). In addition, Day-urban-LVOA showed only a limited similarity
in its variation trend to Day-HNOx-LVOA during the urban air mass period (Fig. S26). This finding
supports the hypothesis that the daytime formation of urban-related OA factors was closely related
to the urban pollutants. Additionally, Day-urban-ELVOA exhibited a positive correlation with
organic vapors (R = 0.65, Fig. S25b), while such a correlation was not observed for Day-urban-
LVOA. It implies that Day-urban-ELVOA may primarily form through gas-particle partitioning
during the urban air mass period.
Day-urban-LVOA was also positively correlated with $NO_3^-$/SIA (Fig. 5c), consistent with the
concurrent enhancement of nitrate and SOA during haze episodes (Ye et al., 2023; Zheng et al.,
2021). During the urban air masses period, nitrate demonstrates a bimodal diurnal variation with
peaks in both the morning and afternoon (Fig. S27), the latter peak likely attributed to $OH+NO_2$
pathway (Yang et al., 2022). Day-urban-LVOA had a significant correlation (R=0.97) with
succinic acid ($C_4H_6O_4$) in the particle phase (Fig. S28), which was previously reported to form via
multiphase reaction during haze episode in megacity (Zhao et al., 2018; Zheng et al., 2021). As
shown in Fig. S29, Day-urban-LVOA also increased with the ratio of the aerosol liquid water
content (ALWC) to $PM_1$, further indicating that aqueous processes in urban plumes played an
important role in its enhancement.
For the aging factors, Day-aged-LVOA and Day-aged-ELVOA exhibited peak concentrations
about 3 hours later (at about 18:00 LT, Fig. 4) than other day factors (15:00 LT). It suggests that the
two aged factors might originate from the photochemical aging processes of preexisting organic
aerosols. To further explore the formation and aging process of these daytime factors, we estimated
their daytime enhancement (Δ). For factors peaked at 15:00 LT, the Δwas estimated as the difference

between the average mass concentration during 00:00-6:00 LT and 12:00-18:00 LT. For factors

peaking at about 18:00 LT, Δ was regarded as the difference between the average mass

concentration during 6:00-12:00 LT and 15:00-21:00 LT, since these factors remained at a relatively

high-level during nighttime probably owing to lower boundary layer height. The Δ Day-aged-

LVOA showed strong positive correlations with Δ Day-HNO$_x$-LVOA (R=0.73), Δ Day-urban-

LVOA (R=0.77), and Δ Day-LNO$_x$-LVOA (R=0.64, Fig. 7a), suggesting that its formation might

be closely associated with the aging processes of these three factors. Similarly, Δ Day-aged-ELVOA

was positively correlated with both Δ Day-LNO$_x$-LVOA (R=0.61), ΔDay-urban-LVOA (R=0.67),

and Δ Day-urban-ELVOA (R=0.73, Fig. 7c). In contrast, we did not observe such correlations

between  Δ Day-aged-ELVOA and ΔDay-HNO$_x$-LVOA (R=0.49, Fig. 7c). It implies that the

formation of Day-aged-ELVOA was likely more influenced by the aging of urban-related factors

and Day-LNO$_x$-LVOA.

**3.3 Comparison with AMS OA**

Adopting PMF analysis to thermogram datasets provides valuable insights into the formation

and aging processes of SOA. However, the representativeness of FIGAERO-OA still requires

evaluation. Figure 8 compares FIGAERO-OA with AMS-OA during two different periods. In

general, FIGAERO-OA could not explain MO-OOA and HOA identified in AMS OA, given that

all thermogram factors had a weak correlation (R=-0.18–0.36) between these two factors (Table S2).

MO-OOA, which had the highest O:C (1.0) among all AMS factors (0.32-1.0) (Cai et al., 2024),

was likely low volatile, meaning that much of this fraction might not have been vaporized during

the heating process. Xu et al. (2019) investigate the volatility of different OA factors using the

TD+AMS method and found that MO-OOA evaporated ~52% at T=175°C. Another TD+AMS field

study in the North China Plain suggested that the volatility of MO-OOA varied with RH levels,

more MO-OOA evaporate at higher RH levels (RH>70, Xu et al., 2021), suggesting that MO-OOA

compounds formed at high RH condition could be higher volatile. During this campaign, the RH

varied from 25% to 92% which likely caused variability in MO-OOA volatility and thus in the

fraction desorbed at 175 °C. This variability might explain the low correlation between MO-OOA

in AMS and all FIGAERO-OA factors. HOA mainly consists of hydrocarbon-like organic

compounds, which could not be detected by the FIGAERO-CIMS. The iodide source of the

FIGAREO-CIMS is selective towards multi-functional organic compounds(Lee et al., 2014),

making it less sensitive to detection hydrocarbon-like species. Ye et al. (2023) preformed
factorization analysis of data obtained from the FIGAERO-CIMS and AMS and suggested that
FIGAERGAERO-CIMS and AMS and suggested that FIGAERO-derived OA factors could not
account for all primary OA components resolved by AMS, including COA, NOA, and HOA. These
findings highlight the need for further investigations into the chemical characteristics of primary
OA to better understand their emission signatures and atmospheric evolution.
BBSOA in AMS-OA had a bimodal diurnal distribution with an afternoon peak (~ 14:00 LT)
and an evening peak (~ 17:00 LT, Fig. S30). The enhancement was more pronounced in the
afternoon (~1.6 to ~ 3.6 µg m$^{-3}$) compared to the evening (~2.9 to ~4.0 µg m$^{-3}$). Thus, we classify
both BBSOA and LOOA as daytime SOA. Six thermogram daytime factors could explain the
majority (82% on average) of daytime SOA with the explained fraction increasing from 78% during
the long-range transport period to 85% during the urban air masses period (Fig. 8 a and b). In both
periods, the summed thermogram daytime factors exhibited a diurnal variation like that of
LOOA+BBSOA (Fig. 8 c and d). Thermogram daytime OA was close to AMS daytime OA in the
morning but fell below AMS OA afternoon. The discrepancy in the afternoon could be related to
the decrease in OA volatility through strong photochemical reactions. Since the heating temperature
of the FIGAERO-CIMS was set at 175°C, compounds of very low volatility might not have been
fully detected. This discrepancy narrowed during the urban air masses, likely owing to the strong
SOA formation through gas-particle partitioning, which increased OA volatility (Cai et al., 2024).
The gap persisted overnight, owing to suppressed vertical mixing under lower boundary layer
conditions.
FIGAERO-OA explained about 13% of BBOA in AMS OA during the campaign and this ratio
remained relatively stable across different periods compared with daytime SOA (Fig. 8 a and b).
Because BBOA is closely tied to local biomass burning activities, air mass variations likely had
only a minor influence on its chemical characteristics. BB-LVOA showed a diurnal pattern similar
to both BBOA in AMS-OA and levoglucosan (Fig. 9a), with an evening peak around 18:30 LT,
confirming their close association with biomass burning emissions. For nighttime chemistry related
factor, both Night-OA (from AMS) and Night-LVOA (from thermograms) increase during the
nighttime, while they did not share a similar diurnal pattern (Fig. 9b). Night-LVOA peaked at about
20:00 LT and decreased after 4:00 LT, while Night-OA peaked later, at about 06:00 LT, and declined
in the morning. It suggested that Night-LVOA identified by FIGAERO-CIMS might not be able to
fully capture the evolution of organic compounds involved in nighttime chemistry, which can
explain 48% of Night-OA in AMS-OA. Given that the majority of organic compounds formed
through the nighttime chemistry were oxygenated and could be detected by FIGAERO CIMS (Wu
et al., 2021), we speculated that the volatility of organic compounds decreased overnight, resulting
that some low volatility organic aerosols would not be fully vaporized during the heating process.
Xu et al. (2019) found that nighttime MO-OOA exhibited lower volatility compared with daytime
MO-OOA, likely due to differences in precursors, formation mechanisms, and meteorological
conditions.

## 4. Conclusion

In this study, we applied a PMF analysis to field thermogram data set measured by the
FIGAERO-CIMS and classified the factors based on their potential formation pathways and
volatility. Based on the PMF analysis to thermograms data sets, six daytime OA factors, a biomass
burning related factor, and nighttime chemistry related factor were identified. The formation of Day-
HNO$_x$-LVOA and Day-LNO$_x$-LVOA was closely related to gas-particle partitioning, while Day-
HNO$_x$-LVOA was observed to be formed with organic vapors under high NO$_x$ condition. The
increase in NO$_x$ concentration might inhibit the production of highly oxygenated compounds (Cai
et al., 2024), which could explain the relatively high volatility of Day-HNO$_x$-LVOA. Two urban
related factors, Day-urban-LVOA and Day-urban-ELVOA, were identified, which only showed a
daytime enhancement in urban plumes. The former might originate from aqueous processes, while
the latter was likely formed through gas-particle partitioning. Our results demonstrated that
photochemical-derived gas-particle partitioning mainly contributed to OA formation in downwind
urban plumes.
Daytime aging processes of organic aerosol were observed and leading to the decrease in
volatility with two aged factors (Day-aged-LVOA and Day-aged-ELVOA) identified. The formation
of Day-aged-LVOA was related to the photochemical aging processes of Day-HNO$_x$-LVOA, Day-
LNO$_x$-LVOA, Day-urban-LVOA, and Day-urban-ELVOA, while Day-aged-ELVOA originates
from the aging processes of Day-LNO$_x$-LVOA, Day-urban-LVOA, and Day-urban-ELVOA. In
general, these six thermogram daytime factors could explain the majority of daytime SOA in AMS

OA, and this ratio increase from 79% during the long-range transport period to 85% during the urban air masses period, probably owing to a higher OA volatility (Cai et al., 2024). While FIGAERO-OA is unable to explain hydrocarbon like OA (HOA) and more oxygenated OA (MOOA), since the FIGAERO-CIMS could not detect hydrocarbon molecules and low volatility organic compounds with a volatilization temperature higher than 170 °C. For biomass-related OA, BB-LVOA could explain about 11%-13% of the BBOA in AMS OA, sharing a similar diurnal pattern, indicating that adopting a PMF analysis to thermogram profile could capture biomass burning events. While Night-LVOA had a different diurnal pattern with Night-OA in AMS OA, implying that this thermogram factor was not unable to represent the evolution of OA during the nighttime.

To our knowledge, existing field studies applying PMF to FIGAERO-CIMS data have primarily focused on the mass concentrations or signal intensities of organic compounds rather than their thermograms. Chen et al. (2020) applied PMF to FIGAERO-CIMS datasets collected in Yorkville, GA, and reported substantial contributions of isoprene- and monoterpene-derived SOA during both daytime and nighttime. Using the same approach, Ye et al. (2023) showed that low-NO-like oxidation pathways played a significant role in SOA formation in urban environments. However, these PMF analyses did not provide volatility information, which limits our ability to fully understand the formation mechanisms and aging processes of OA. Lee et al. (2020) demonstrated that combining molecular-level composition measurements with volatility information enables the resolution of organic aerosol formation and aging pathways in the atmosphere, providing direct constraints on how oxidation processes alter both chemical functionality and volatility during aerosol evolution. Buchholz et al. (2020) performed PMF analysis on FIGAERO-CIMS thermogram datasets in laboratory experiments and demonstrated that both OA volatility and composition varied with relative humidity. Nevertheless, applications of thermogram-based PMF to ambient field measurements remain scarce.

Our results show that applying PMF directly to thermogram profiles from field observations yields additional and valuable volatility information that is not accessible from traditional mass- or signal-based PMF analyses. This added dimension is particularly useful for OA source apportionment. Along with PMF analysis of AMS or ACSM data, it can provide crucial information in understanding the formation and aging processes of OA. Using this method, we found that the daytime atmospheric evolution of SOA involved gas–particle partitioning, aqueous-phase reactions,

and photochemical aging, highlighting the complexity of daytime SOA formation. Moreover, SOA
volatility was strongly dependent on its formation pathways. variations in $NO_x$ not only influenced
atmospheric oxidation but also modified SOA volatility by altering formation mechanisms.
Nevertheless, further investigations are required to clarify the role of urban plumes in shaping SOA
formation and its physicochemical properties.


*Data availability.* Data from the measurements are available at 10.6084/m9.figshare.30155584

*Supplement.* The supplement related to this article is available online at xxx.

*Author contributions.* M.C., W. H., and B.Y. designed the research. M.C., B.Y., W.H., Y.C., S. H.,
S.Y., W.C., Y. P., and J.Z. performed the measurements. M. C., B.Y., W.H., Y.C., S. H., Z. D., and
D. C. analyzed the data. M. C., W.H. and B.Y. wrote the paper with contributions from all co-authors.

*Competing interests.* The authors declare that they have no conflict of interest.

*Financial support.* This work was supported by Guangdong Basic and Applied Basic Research
Foundation (grant nos. 2024A1515030221, 2023A1515012240), National Natural Science
Foundation of China (grant no. 42305123, 42375105), Science and Technology Projects in
Guangzhou (grant no. 2025A04J4493), the Key Innovation Team of Guangdong Meteorological
Bureau(No. GRMCTD202506-ZD06), and Central Public interest Scientific Institution Basal
Research Fund of South China Institute of Environmental Sciences, MEE (grant no. PM-zx097-

671 202506-214).

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

**Table 1.** The average volatility ($\log_{10} \overline{C^*}$), $T_{max}$, signal-weighted average values of elemental
composition, carbon oxidation state ($\overline{OS_c}$), H:C, O:C, N:C for all FIGAERO-OA factors. The
estimation of $\overline{OS_c}$ can be found in Section S2. The volatility of each FIGAERO-OA factor was
estimated based on their corresponding $T_{max}$ using eq. (8) and (9).

| | $\log_{10} \overline{C^*}$ ($\mu g\ m^{-3}$) | $T_{max}$ (°C) | Average elemental composition | $\overline{OS_c}$ | H:C | O:C | N:C |
|---|---|---|---|---|---|---|---|
| Day-HNO$_x$-LVOA | -0.98 | 84.52 | $C_{7.37}H_{10.51}O_{4.99}N_{0.36}$ | -0.01 | 1.37 | 0.75 | 0.06 |
| Day-LNO$_x$-LVOA | -2.71 | 103.29 | $C_{6.52}H_{8.77}O_{4.54}N_{0.22}$ | 0.18 | 1.35 | 0.80 | 0.04 |
| Day-aged-LVOA | -2.02 | 95.53 | $C_{6.35}H_{8.75}O_{5.13}N_{0.21}$ | 0.35 | 1.42 | 0.91 | 0.04 |
| Day-aged-ELVOA | -4.80 | 126.65 | $C_{5.22}H_{7.36}O_{4.20}N_{0.16}$ | 0.40 | 1.55 | 1.00 | 0.03 |
| Day-urban-LVOA | -0.90 | 83.03 | $C_{6.50}H_{9.27}O_{4.71}N_{0.24}$ | 0.08 | 1.43 | 0.80 | 0.04 |
| Day-urban-ELVOA | -7.18 | 153.22 | $C_{6.57}H_{8.54}O_{4.61}N_{0.24}$ | 0.26 | 1.35 | 0.84 | 0.05 |
| BB-LVOA | -2.36 | 99.39 | $C_{6.72}H_{9.78}O_{4.61}N_{0.26}$ | -0.08 | 1.47 | 0.74 | 0.04 |
| Night-LVOA | -2.02 | 95.53 | $C_{7.69}H_{11.04}O_{5.19}N_{0.47}$ | -0.09 | 1.47 | 0.77 | 0.07 |


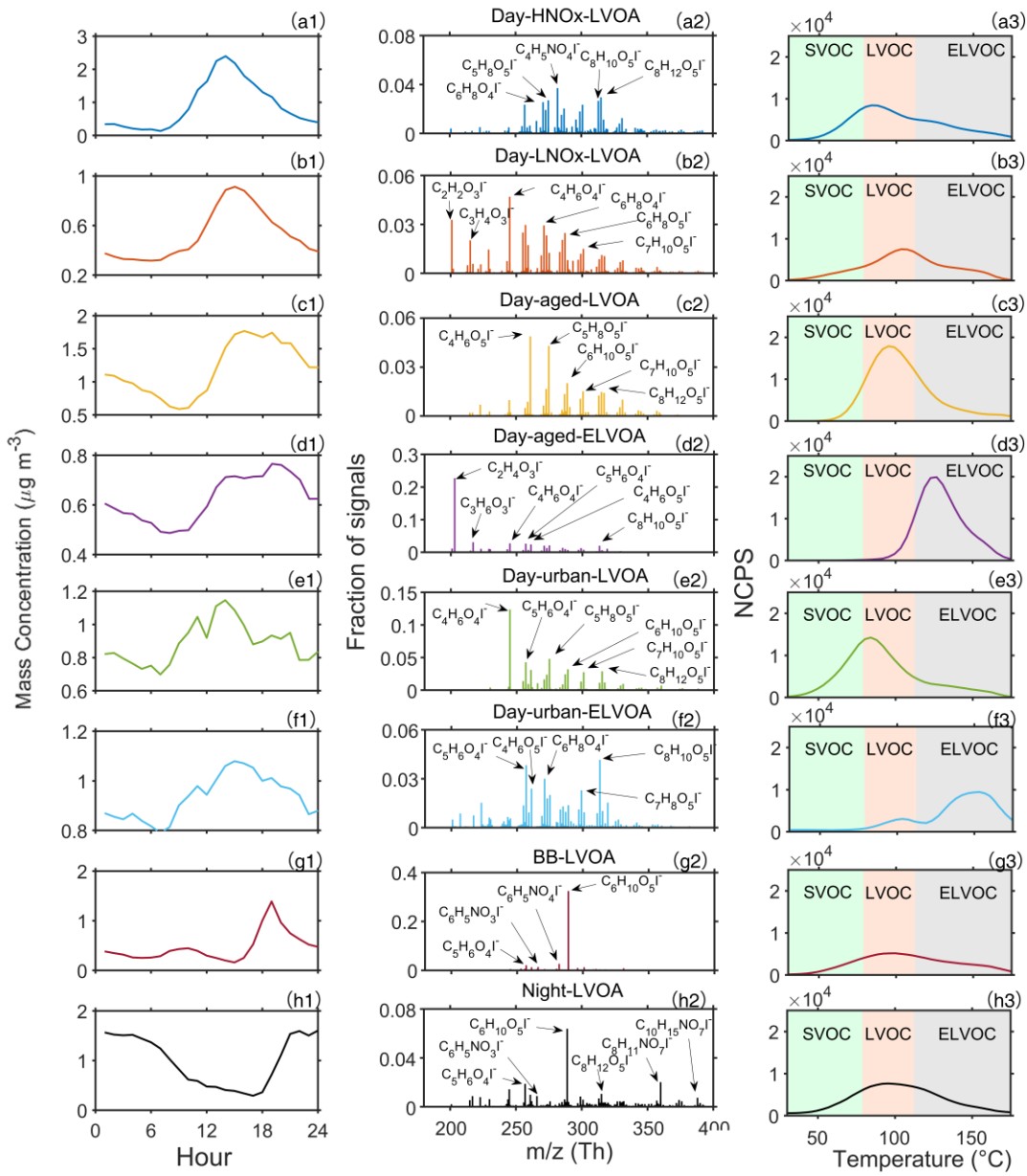


**Figure 1.** Diurnal variation (a1 to h1), mass spectra (a2 to h2), and thermograms (a3 to h3) of
FIGAERO-OA factors.

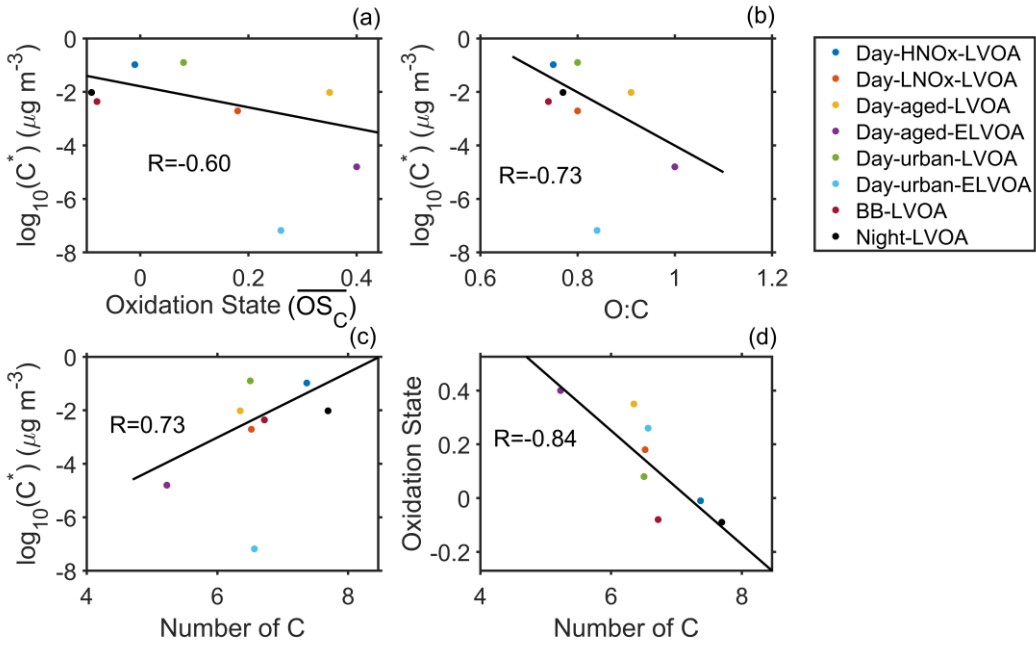


**Figure 2.** The average volatility of FIGAERO-OA factor vs. (a) oxidation state ($\overline{OS_c}$,), (b) O:C ,
and (c) number of carbons and (d) Number of carbons vs. $\overline{OS_c}$ of thermogram factor. Day-urban-
ELVOA is excluded in the estimation of R.


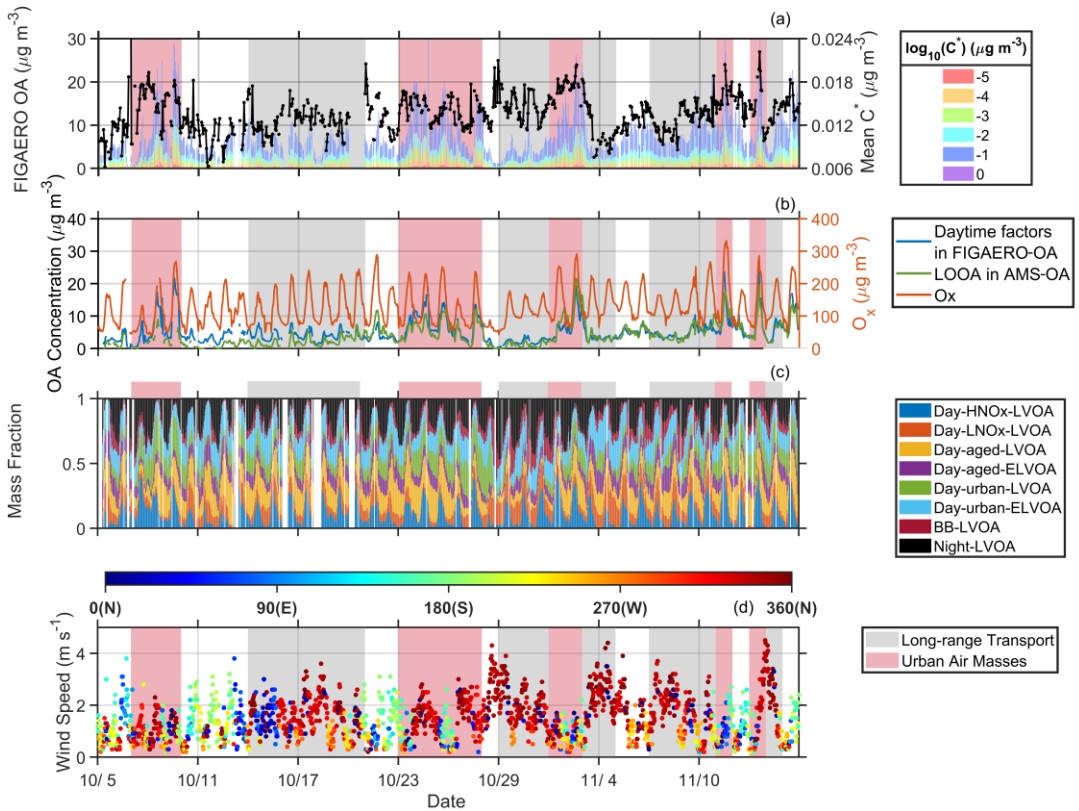


**Figure 3.** Time series of (a) volatility (presented in a range from $10^{-5}$ to $10^0$ μg m$^{-3}$) of FIGAERO-
OA and mean $C^*$, (b) daytime factors (Day-HNO$_x$-LVOA, Day-LNO$_x$-LVOA, Day-aged-LVOA,
Day-aged-ELVOA, Urban LVOA, and Day-urban-ELVOA) in FIGAERO-OA and LOOA factor
from PMF analysis of SP-AMS data, (c) mass fraction of eight FIGAERO-OA factors, and (d) wind
speed and wind direction.


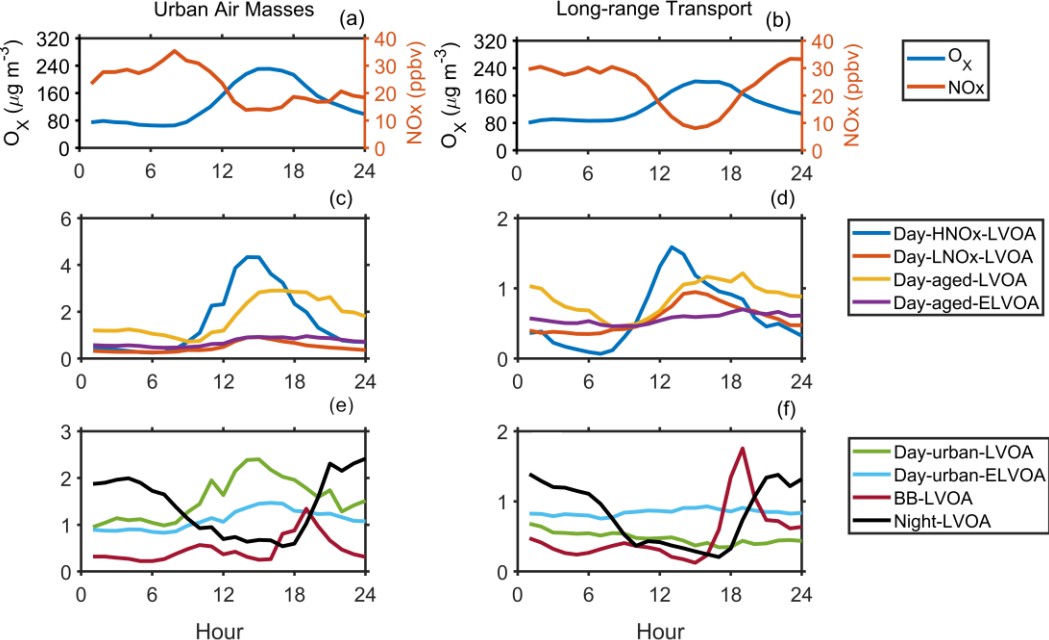


**Figure 4.** The average diurnal variation of $O_x$, $NO_x$, and mass concentration of eight thermogram
factors during the long-range transport (a, c, and e) and urban air masses (b, d, and f) period.


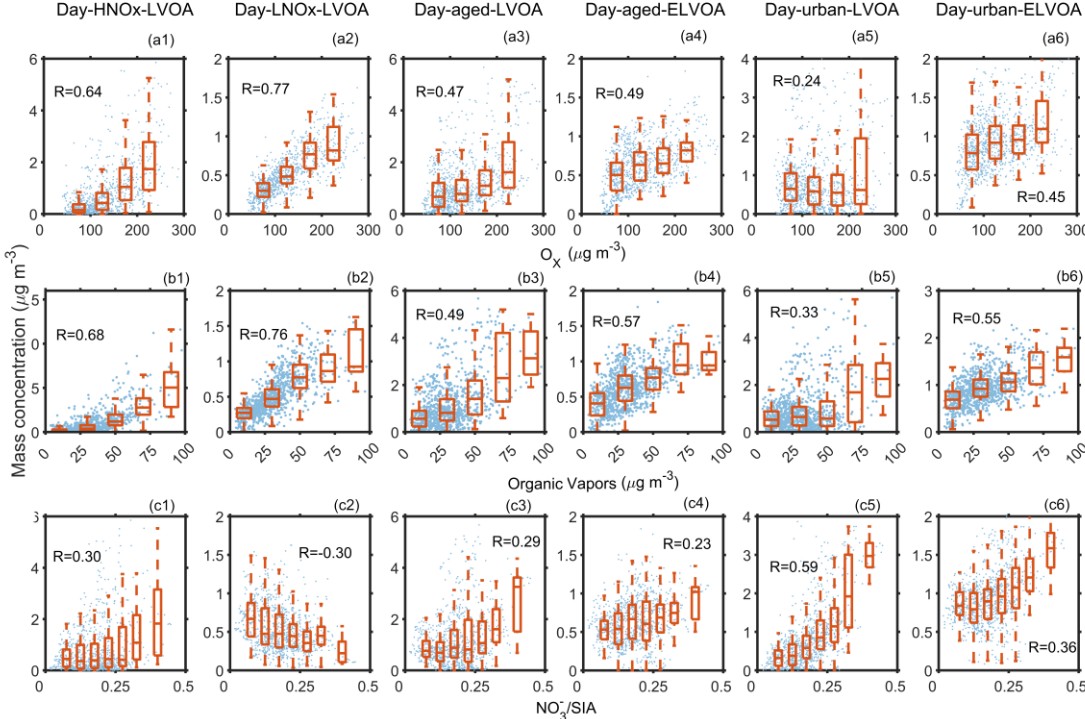

**Figure 5.** Relationship between the mass concentration of six daytime thermogram factors and
(a1-6) $O_x$, (b1-6) organic vapors, (c1-6) nitrate fraction in secondary inorganic aerosol (SIA), and
(d1-6) sulfate fraction in SIA measured by the FIGAERO-CIMS. The organic vapors are the sum
of organic compounds in the gas-phase measured by the FIGAERO-CIMS.

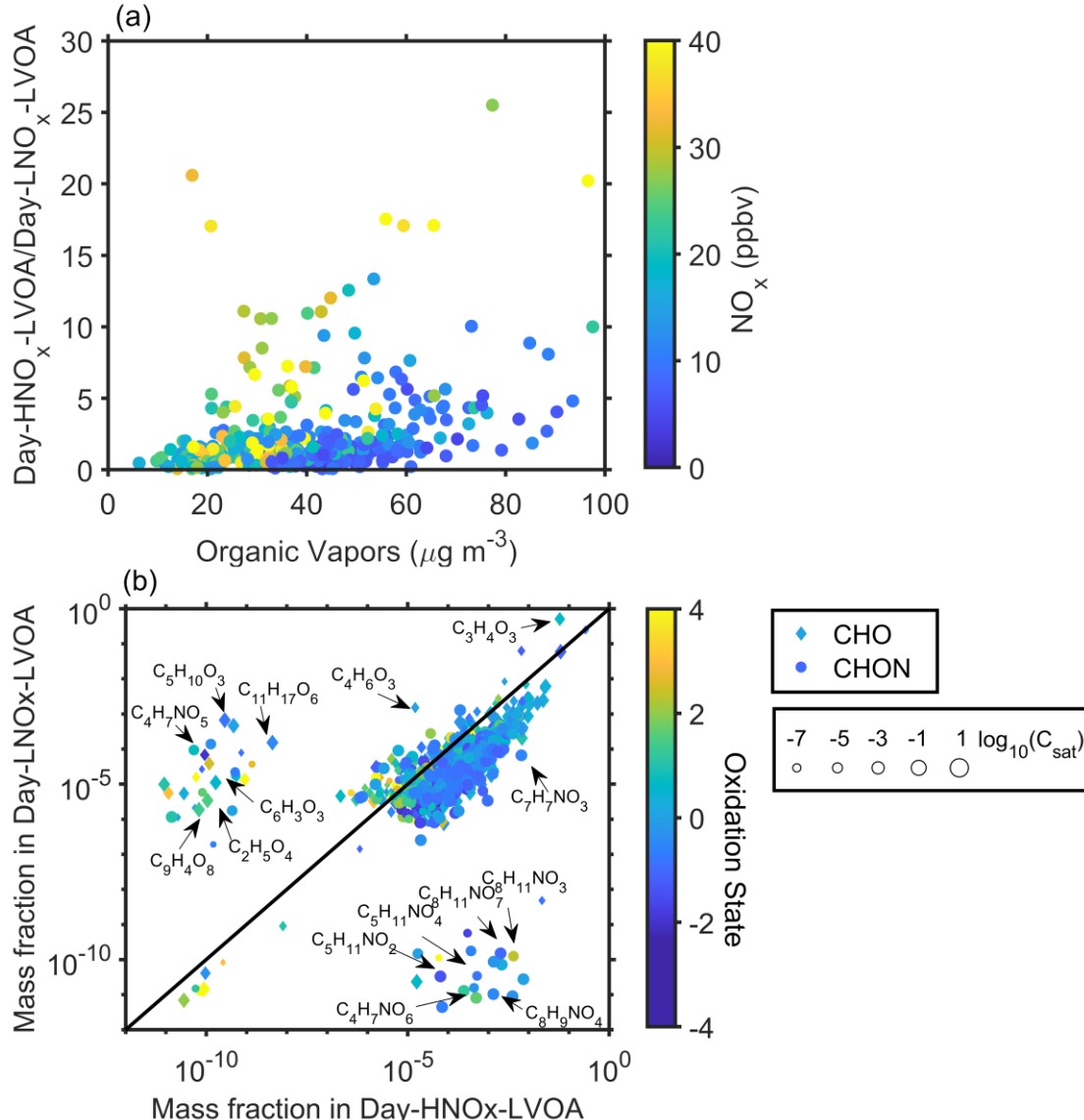

**Figure 6.** (a) Correlation between organic vapors and the ratio of Day-HNO$_x$-LVOA to Day-LNO$_x$-LVOA. (b) Scatterplots of mass fraction of different species in Day-HNO$_x$-LVOA and Day-LNO$_x$-LVOA. The color of dots in panel (a) denotes the corresponding NO$_x$. The shape, size, and color of markers in panel (b) represents the class of species, volatility, and $\overline{OS_C}$, respectively.


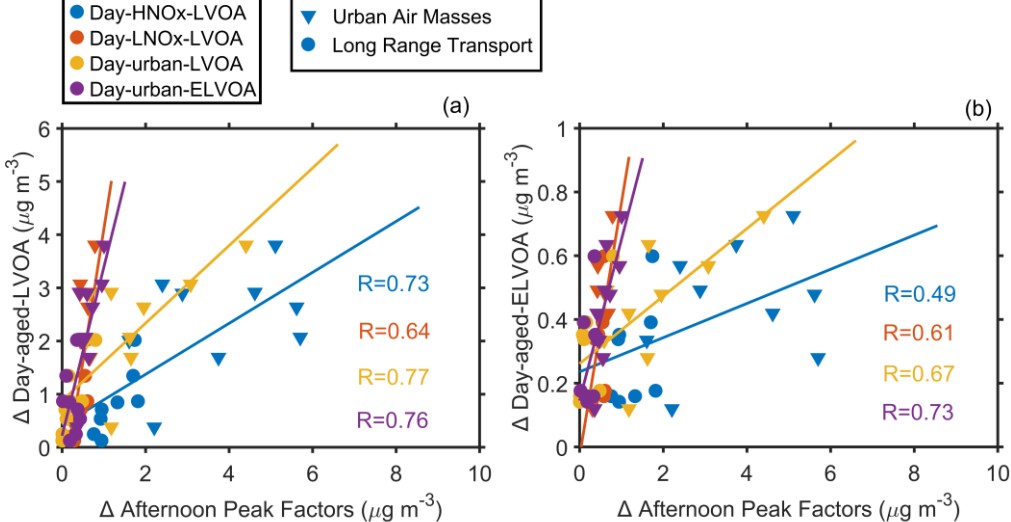


**Figure 7.** Correlation between the enhancement of (a) Day-aged-LVOA and afternoon peak factors
and (b) Day-aged-ELVOA and afternoon peak factors. Afternoon peak factors include Day-HNO$_x$-
LVOA, Day-LNO$_x$-LVOA, Day-urban-LVOA, and Day-urban-ELVOA. For afternoon peak factors,
the enhancement ($\Delta$) was regarded as the average mass concentration during 00:00-6:00 LT and
12:00-18:00 LT. For Day-aged-LVOA and Day-aged-ELVOA, the enhancement ($\Delta$) was estimated
as the difference between average mass concentration during 00:00-6:00 LT and 12:00-18:00 LT.

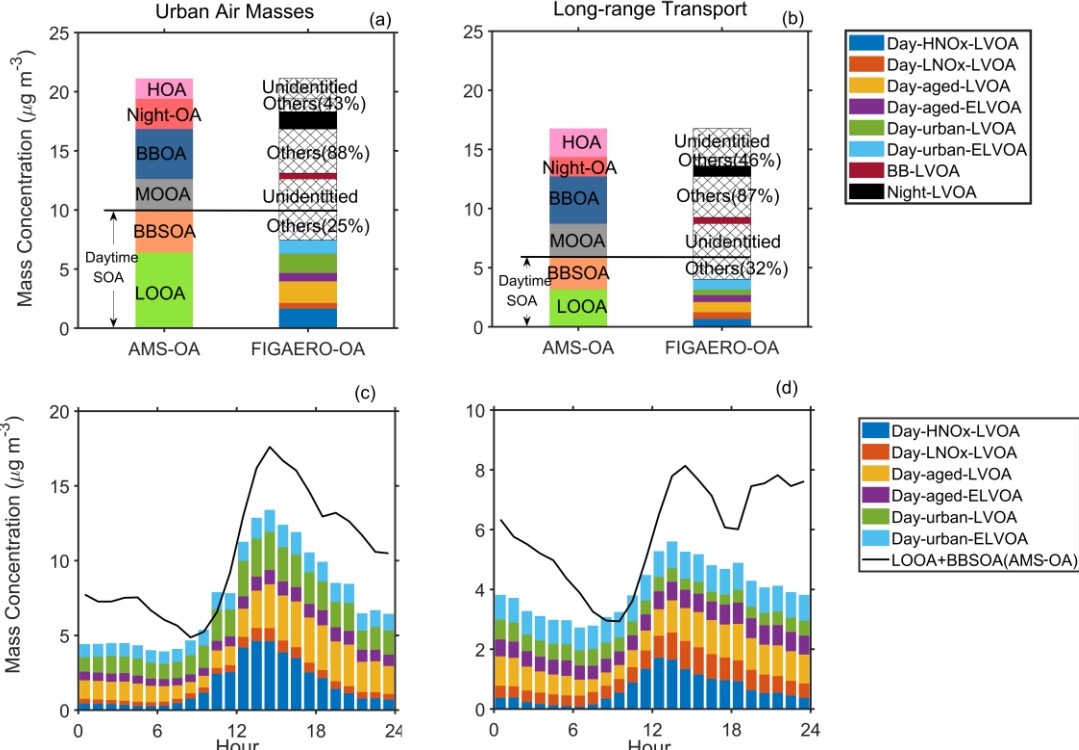

**Figure 8.** Comparison of the average mass concentration (a and b) and diurnal variation (c and d)
of AMS-OA and FIGAERO-OA during long-range transport and urban air masses period.


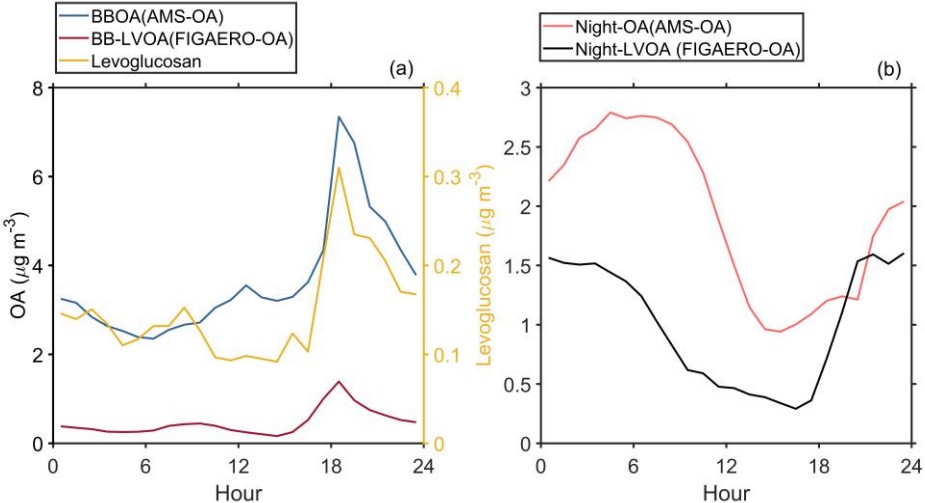


**Figure 9. (a)** Diurnal variation of BBOA from AMS, BB-LVOA and levoglucosan from FIGAERO-
CIMS;(b) Diurnal variation of Night-OA from AMS, and Night-LVOA from FIGAERO-CIMS.