# Peer review of "New insight into the formation and aging processes of organic"

_EGUsphere, 2025_

## Author Comment (AC1)

The paper provides a comprehensive analysis of SOA formation and aging processes in the PRD region, using advanced measurements from a FIGAERO-CIMS coupled with PMF analysis. The study identifies and characterizes different SOA factors based on their volatility and formation pathways. The results highlight the significant role of gas-particle partitioning and photochemical aging in SOA formation, with variations driven by environmental factors such as $NO_x$ levels. The authors also compare these findings with data from AMS and discuss the limitations of FIGAERO-CIMS in detecting certain OA components. This manuscript is suitable for publication in ACP and I recommend it for publication after the following comments have been addressed.

1. The author mentioned that six daytime FIGAERO factors were positively correlated with LOOA in AMS OA. I wonder whether the relationship between FIGAERO factors and LOOA varies across different periods.

Reply: We appreciate the reviewer for this valuable suggestion. We examined the relationship between FIGAERO factors and LOOA during urban both air massed and long-range transport period. The sum of six daytime FIGAERO factors showed a positive relationship (R=0.80 and 0.76, respectively) with LOOA during both periods. However, the slope (0.81) of the linear regression during the urban air masses period was higher than that (0.58) during the long-range transport period, indicating that a higher fraction of LOOA could be detected by the FIGAERO-CIMS during urban air masses period. This difference could be related to the difference in OA volatility. According to Cai et al. (2024), the volatility of OA was higher during the urban air masses period.

We added some discussion in line 415-421,

"**Note that the sum of six daytime FIGAERO factors showed a positive relationship (R=0.80 and 0.76, respectively) with LOOA during both periods (Fig. S20). However, the slope (0.81) of the linear regression during the urban air masses period was higher than that (0.58) during the long-range transport period, indicating that a higher fraction of LOOA could be detected by the FIGAERO-CIMS during urban air masses period. This difference could be related to the discrepancy in OA volatility. According to Cai et al. (2024), the volatility of OA was higher during the urban air masses period.**"

[Figure]

**Figure S20.** Correlation between the sum of six daytime FIGAERO-OA factors and LOOA during different periods.

2. The calibration experiment regarding the relationship between Tmax and saturation vapor concentration is important. Could the authors provide more details about this calibration experiment in the main text? Specifically, why were the fitting parameters chosen?

Reply: We appreciate the reviewer for this valuable suggestion. We have added some introduction about the calibration experiment and the selection of fitting parameters in line 221-232:

**"The fitting parameters of a and b were calibrated by a series of polyethylene glycol (PEG 5-8) compounds before the campaign. PEG standards (dissolved in acetonitrile) were atomized using a homemade atomizer, and the resulting particles were size-classified by a differential mobility analyzer (DMA; model 3081L, TSI Inc.) to target diameters of 100 and 200 nm. The size-selected particles were then split into two flows: one directed go to a CPC (3775, TSI) for the measurements of number concentration, and the other to the FIGAERO-CIMS particle inlet. The collected mass by CIMS was calculated based on the particle diameter, number concentration, FIGAERO-CIMS inlet flow rate, and collection time. The details of the calibration experiments and selection of fitting coefficients (a and b) can be found in table S1 and Cai et al. (2024). In this study, the fitting parameters (a=-0.206 and b=3.732) were chosen, as the mass loading (407 ng) and diameter (200 nm) are closest to the ambient samples, since the collected mass loading centered at about 620 ng and the particle volume size distribution (PVSD) centered at about 400 nm (Cai et al., 2024)."**

3. The study finds discrepancies between nighttime SOA measured by AMS and that characterized by FIGAERO-CIMS, suggesting that the nighttime processes may not be fully captured by the FIGAERO-CIMS thermogram data. Is there any evidence about the low volatility of nighttime

OA?

Reply: We appreciated the reviewer for this valuable suggestion. It is indeed challenging for FIGAERO-CIMS to fully capture the relatively low volatile fraction of OA, owing to the "relatively low heating temperature (~175℃)". During the same campaign, a thermodenuder (TD) coupled with an AMS was also deployed to characterize OA volatility. The TD operated at temperatures up to ~270 °C, enabling the detection of substantially lower-volatility OA. We are currently preparing our next manuscript focusing the volatility of nighttime OA and a comparison between the FIGAERO-CIMS and TD-AMS approaches. As the TD-AMS dataset is still under analysis, we are unable to include those results in the present manuscript.

Previous studies also support the likelihood of lower nighttime OA volatility. For example, Xu et al. (2019) found that volatility of MO-OOA was lower at nighttime than daytime, likely due to differences in precursor emissions, formation pathways, and meteorological conditions. In addition, organic nitrates have lower volatility than hydroxylated products with the same carbon number(Donahue et al., 2011; Ren et al., 2022). Taken together, these findings suggest that nighttime chemistry, which produces a higher fraction of organic nitrates, may generate OA with substantially lower volatility. (Kiendler-Scharr et al., 2016)"

We added some discussion to the revised manuscript in line 591-595,

"**Xu et al. (2019) found that nighttime MO-OOA exhibited lower volatility compared with daytime MO-OOA, likely due to differences in precursors, formation mechanisms, and meteorological conditions. Moreover, organic nitrates generally have lower volatility than hydroxylated species with the same carbon number (Donahue et al., 2011; Ren et al., 2022). It suggested that a higher fraction of nighttime organic nitrates could lead to lower OA volatility (Kiendler-Scharr et al., 2016).**"

4. The authors suggest that an increase in NOx levels could enhance the volatility of SOA. I recommend that the authors compare this finding with other studies on the impact of NOx on OA volatility.

Reply: We appreciate the reviewer for this valuable suggestion. To our current knowledge, most studies investigating the influence of $NO_x$ on the volatility of OA have been conducted under controlled laboratory conditions, while field-based evidence remains limited. D'ambro et al. (2017) investigate the molecular composition and volatility of isoprene derived SOA under high and low $NO_x$ condition in an environmental simulation chamber. Their results showed that SOA exhibited lower volatility under high-$NO_x$ conditions, corresponding to a greater contribution of organic nitrates. However, the experimental conditions were restricted to two scenarios: high $NO_x$ and low $NO_x$. Furthermore, in the high-$NO_x$ experiments, the NO input

was 20 ppb, without accounting for the nonlinear dependence of SOA formation pathways on $NO_x$ concentrations (Pye et al., 2019).

Xu et al. (2014) further investigated the variation of SOA volatility over a wide range of $NO_x$ levels (<1 ppb to 738.1 ppb) in a series of chamber experiments. They found that both SOA volatility and oxidation state exhibited a nonlinear response to $NO_x$. SOA volatility decreases with increasing $NO_x$ level when the ratio of initial NO to isoprene was lower than 3. At higher $NO_x$ level, higher volatile SOA was produced, probably owing to the more competitive $RO_2$+NO pathway. This study highlights the important nonlinear impacts of $NO_x$ concentrations on SOA formation and volatility. More field measurements were needed to investigate these effects in the ambient environment.

We added some discussion to the revised manuscript in line 482-486,

"**Xu et al. (2014) found that both SOA volatility and oxidation state exhibited a nonlinear response to $NO_x$ in a series of chamber environment. SOA volatility decreases with increasing $NO_x$ level when the ratio of initial NO to isoprene was lower than 3. At higher $NO_x$ level, higher volatile SOA was produced, probably owing to the more competitive $RO_2$+NO pathway**."

5. The paper finds that FIGAERO-OA cannot explain MO-OOA and HOA in AMS, but it does not further analyze the reasons. For MO-OOA, it is unclear whether it is a "very low-volatility species not desorbed by heating". For HOA, the undetection may be due to the low response efficiency of the ionization method ($I^-$ reagent) in FIGAERO-CIMS, which weakens the discussion on the data complementarity of the two instruments.

Reply: We would like to thank the reviewer for this valuable suggestion. We acknowledge that capturing the full spectrum of OA using FIGAERO-CIMS remains challenging. One limitation arises from the relatively low maximum heating temperature (~175 °C), which prevents full desorption of low-volatility OA. Xu et al. (2019) investigate the volatility of different OA factors using the TD+AMS method and found that MO-OOA evaporated ~52% at T=175°C. Another TD+AMS field study in the North China Plain suggested that the volatility of MO-OOA varied with RH levels(Xu et al., 2021), more MO-OOA evaporate at higher RH levels (RH>70%). During this campaign, the RH varied from 25% to 92% which likely caused variability in MO-OOA volatility and thus in the fraction desorbed at 175 °C. This variability might explain the low correlation between MO-OOA in AMS and all FIGAERO-OA factors.

In addition, the iodide source of the FIGAREO-CIMS is selective towards multi-functional organic compounds(Lee et al., 2014), making it less sensitive to detection hydrocarbon-like species. Ye et al. (2023) preformed factorization analysis of data obtained from the FIGAERO-

CIMS and AMS and suggested that FIGAERO-derived OA factors could not account for all primary OA components resolved by AMS, including COA, NOA, and HOA. These findings highlight the need for further investigations into the chemical characteristics of primary OA to better understand their emission signatures and atmospheric evolution.

We added some discussion to the revised manuscript in line 545-552,

"**Xu et al. (2019) investigate the volatility of different OA factors using the TD+AMS method and found that MO-OOA evaporated ~52% at T=175°C. Another TD+AMS field study in the North China Plain suggested that the volatility of MO-OOA varied with RH levels, more MO-OOA evaporate at higher RH levels (RH>70, Xu et al., 2021), suggesting that MO-OOA compounds formed at high RH condition could be higher volatile. During this campaign, the RH varied from 25% to 92% which likely caused variability in MO-OOA volatility and thus in the fraction desorbed at 175 °C. This variability might explain the low correlation between MO-OOA in AMS and all FIGAERO-OA factors.**"

and in line 553-560,

"**The iodide source of the FIGAREO-CIMS is selective towards multi-functional organic compounds(Lee et al., 2014), making it less sensitive to detection hydrocarbon-like species. Ye et al. (2023) preformed factorization analysis of data obtained from the FIGAERO-CIMS and AMS and suggested that FIGAERO-derived OA factors could not account for all primary OA components resolved by AMS, including COA, NOA, and HOA. These findings highlight the need for further investigations into the chemical characteristics of primary OA to better understand their emission signatures and atmospheric evolution.**"

6.  Figure 2 a: please add "" to the x-axis label.

    Reply: The character in the "" is missing. We thought it could be "$\overline{OS_c}$" and revised figure 2 a.

7.  Table 1: It should be "average elemental composition" in the heading.

    Reply: It has been revised.

8.  Eq (5) and (6) are missing in the main text. This is presumably a typesetting omission.

    Reply: It has been revised.

**Reference:**

Cai, M., Ye, C., Yuan, B., Huang, S., Zheng, E., Yang, S., Wang, Z., Lin, Y., Li, T., Hu, W., Chen, W., Song, Q., Li, W., Peng, Y., Liang, B., Sun, Q., Zhao, J., Chen, D., Sun, J., Yang, Z., and Shao, M.: Enhanced daytime secondary aerosol formation driven by gas–particle partitioning in downwind urban plumes, Atmos. Chem. Phys., 24, 13065-13079, 10.5194/acp-24-13065-2024, 2024.

D'Ambro, E. L., Lee, B. H., Liu, J., Shilling, J. E., Gaston, C. J., Lopez-Hilfiker, F. D., Schobesberger, S., Zaveri, R. A., Mohr, C., Lutz, A., Zhang, Z., Gold, A., Surratt, J. D., Rivera-Rios, J. C., Keutsch, F. N., and Thornton, J. A.: Molecular composition and volatility of isoprene photochemical oxidation secondary organic aerosol under low- and high-NOx conditions, Atmos. Chem. Phys., 17, 159-174, 10.5194/acp-17-159-2017, 2017.

Donahue, N. M., Epstein, S. A., Pandis, S. N., and Robinson, A. L.: A two-dimensional volatility basis set: 1. organic-aerosol mixing thermodynamics, Atmos. Chem. Phys., 11, 3303-3318, 10.5194/acp-11-3303-2011, 2011.

Kiendler-Scharr, A., Mensah, A. A., Friese, E., Topping, D., Nemitz, E., Prevot, A. S. H., Äijälä, M., Allan, J., Canonaco, F., Canagaratna, M., Carbone, S., Crippa, M., Dall Osto, M., Day, D. A., De Carlo, P., Di Marco, C. F., Elbern, H., Eriksson, A., Freney, E., Hao, L., Herrmann, H., Hildebrandt, L., Hillamo, R., Jimenez, J. L., Laaksonen, A., McFiggans, G., Mohr, C., O'Dowd, C., Otjes, R., Ovadnevaite, J., Pandis, S. N., Poulain, L., Schlag, P., Sellegri, K., Swietlicki, E., Tiitta, P., Vermeulen, A., Wahner, A., Worsnop, D., and Wu, H. C.: Ubiquity of organic nitrates from nighttime chemistry in the European submicron aerosol, Geophysical Research Letters, 43, 7735-7744, https://doi.org/10.1002/2016GL069239, 2016.

Lee, B. H., Lopez-Hilfiker, F. D., Mohr, C., Kurtén, T., Worsnop, D. R., and Thornton, J. A.: An Iodide-Adduct High-Resolution Time-of-Flight Chemical-Ionization Mass Spectrometer: Application to Atmospheric Inorganic and Organic Compounds, Environmental Science & Technology, 48, 6309-6317, 10.1021/es500362a, 2014.

Pye, H. O. T., D'Ambro, E. L., Lee, B. H., Schobesberger, S., Takeuchi, M., Zhao, Y., Lopez-Hilfiker, F., Liu, J., Shilling, J. E., Xing, J., Mathur, R., Middlebrook, A. M., Liao, J., Welti, A., Graus, M., Warneke, C., de Gouw, J. A., Holloway, J. S., Ryerson, T. B., Pollack, I. B., and Thornton, J. A.: Anthropogenic enhancements to production of highly oxygenated molecules from autoxidation, Proceedings of the National Academy of Sciences, 116, 6641-6646, 10.1073/pnas.1810774116, 2019.

Ren, S., Yao, L., Wang, Y., Yang, G., Liu, Y., Li, Y., Lu, Y., Wang, L., and Wang, L.: Volatility parameterization of ambient organic aerosols at a rural site of the North China Plain, Atmos. Chem. Phys., 22, 9283-9297, 10.5194/acp-22-9283-2022, 2022.

Xu, L., Kollman, M. S., Song, C., Shilling, J. E., and Ng, N. L.: Effects of NOx on the Volatility of Secondary Organic Aerosol from Isoprene Photooxidation, Environmental Science & Technology, 48, 2253-2262, 10.1021/es404842g, 2014.

Xu, W., Chen, C., Qiu, Y., Li, Y., Zhang, Z., Karnezi, E., Pandis, S. N., Xie, C., Li, Z., Sun, J., Ma, N., Xu, W., Fu, P., Wang, Z., Zhu, J., Worsnop, D. R., Ng, N. L., and Sun, Y.: Organic aerosol volatility and viscosity in the North China Plain: contrast between summer and winter, Atmos. Chem. Phys., 21, 5463-5476, 10.5194/acp-21-5463-2021, 2021.

Xu, W., Xie, C., Karnezi, E., Zhang, Q., Wang, J., Pandis, S. N., Ge, X., Zhang, J., An, J., Wang, Q., Zhao, J., Du, W., Qiu, Y., Zhou, W., He, Y., Li, Y., Li, J., Fu, P., Wang, Z., Worsnop, D. R.,

and Sun, Y.: Summertime aerosol volatility measurements in Beijing, China, Atmos. Chem. Phys., 19, 10205-10216, 10.5194/acp-19-10205-2019, 2019.

Ye, C., Liu, Y., Yuan, B., Wang, Z., Lin, Y., Hu, W., Chen, W., Li, T., Song, W., Wang, X., Lv, D., Gu, D., and Shao, M.: Low-NO-like Oxidation Pathway Makes a Significant Contribution to Secondary Organic Aerosol in Polluted Urban Air, Environmental Science & Technology, 10.1021/acs.est.3c01055, 2023.

---

## Author Comment (AC2)

This manuscript presents source apportionment of organic aerosol (OA) measured by a FIGAERO-CIMS at a coastal downwind receptor site and resolves eight organic aerosol factors using PMF, followed by a comparison with HR-AMS measurements. Eight factors include six daytime chemistry related factors, a biomass burning related factor (BB-LVOA), and a nighttime chemistry related factor (Night-LVOA). It was also found that increasing NO$x$ levels mainly affected SOA formation via gas-particle partitioning, suppressing the formation of low-volatile organic vapors. Besides, two aged OA factors (Day-aged-LVOA and Day-aged-ELVOA) were mainly attributed to daytime photochemical aging of pre-existing OA.

The topic is scientifically relevant, particularly given the increasing interest in linking OA volatility, oxidation state, and formation pathways. The dataset is valuable, and the thermogram-based OA classification is potentially insightful. However, the attribution of the resolved factors (e.g., High-NOx LVOA, Urban-LVOA, Aged-LVOA) remains insufficiently supported by the current evidence. The manuscript would benefit from clearer methodological justification, more cautious interpretation of factor identities, and additional analyses to better constrain potential chemical and meteorological influences on factor behavior.

1. The authors should clearly state what scientific insights are genuinely new compared with previous FIGAERO-CIMS PMF studies.

   Reply: We appreciate the reviewer for this valuable suggestion. To our knowledge, existing field studies applying PMF to FIGAERO-CIMS data have primarily focused on the mass concentrations or signal intensities of organic compounds rather than their thermograms. Chen et al. (2020) applied PMF to FIGAERO-CIMS datasets collected in Yorkville, GA, and reported substantial contributions of isoprene- and monoterpene-derived SOA during both daytime and nighttime. Using the same approach, Ye et al. (2023) showed that low-NO-like oxidation pathways played a significant role in SOA formation in urban environments. However, these PMF analyses did not provide volatility information, which limits our ability to fully understand the formation mechanisms and aging processes of OA.

   Buchholz et al. (2020) performed PMF analysis on FIGAERO-CIMS thermogram datasets in laboratory experiments and demonstrated that both OA volatility and composition varied with relative humidity. Nevertheless, applications of thermogram-based PMF to ambient field measurements remain scarce.

   In this work, we derive the volatility associated with each OA source and type. Our results show that relatively high-volatility SOA was produced through gas–particle partitioning under elevated NO$_x$ conditions, likely due to the suppressing of NO$_x$ on the formation of highly oxidized organic compounds.

We added some discussion to the revised manuscript in line 625-639,

"**To our knowledge, existing field studies applying PMF to FIGAERO-CIMS data have primarily focused on the mass concentrations or signal intensities of organic compounds rather than their thermograms. Chen et al. (2020) applied PMF to FIGAERO-CIMS datasets collected in Yorkville, GA, and reported substantial contributions of isoprene- and monoterpene-derived SOA during both daytime and nighttime. Using the same approach, Ye et al. (2023) showed that low-NO-like oxidation pathways played a significant role in SOA formation in urban environments. However, these PMF analyses did not provide volatility information, which limits our ability to fully understand the formation mechanisms and aging processes of OA. Lee et al. (2020) demonstrated that combining molecular-level composition measurements with volatility information enables the resolution of organic aerosol formation and aging pathways in the atmosphere, providing direct constraints on how oxidation processes alter both chemical functionality and volatility during aerosol evolution. Buchholz et al. (2020) performed PMF analysis on FIGAERO-CIMS thermogram datasets in laboratory experiments and demonstrated that both OA volatility and composition varied with relative humidity. Nevertheless, applications of thermogram-based PMF to ambient field measurements remain scarce**.

Our results show that applying PMF directly to thermogram profiles from field observations yields additional and valuable volatility information that is not accessible from traditional mass- or signal-based PMF analyses. This added dimension is particularly useful for OA source apportionment."

2. In lines 192-193, the thermogram matrix was split into three segments for PMF due to computational limitations, but the implications for factor consistency and rotational ambiguity were not discussed. A justification and uncertainty evaluation are needed.

Reply: We appreciate the reviewer for this valuable suggestion. We note that a detailed discussion on the justification, factor consistency, and uncertainty associated with the PMF analysis was originally provided in the SI. We have now added a concise summary of this discussion to the main text in line 194-200,

"**An eight-factor solution was selected for each part based on $Q/Q_{exp}$ behavior and factor interpretability (Fig. S3 to S6). To assess factor consistency, the mass spectra of resolved factors were compared across different parts, showing strong correlations (R>0.9) for the each factor (Fig. S7 and S8). Weaker correlations during the early campaign period (2 to 5 October) likely reflect changes in**

dominant OA sources under different meteorological conditions (Fig. S8 and S9). After excluding this period, consistent factor profiles were obtained and combined for further analysis. Detailed evaluations are provided in the Section S1."

We also revised section S1 in the SI as follows,

"**Section S1. Dataset separation and source apportionment for FIGAERO-OA**

We first divided the dataset into 3 parts: part 1 (18690×1028), from 2 to 15 October; part 2 (18970×1028), from 16 to 30 October; part 3 (21840×1028), from 31 October to 16 November. In general, a significant change in $Q/Q_{exp}$ was observed by increasing factors from 2 to 4 (Fig. S3 to S6). After investigating different solutions with factor number from 2 to 10 with fPeak varying between -1 and 1, an 8-factor solution was selected based on the best performance by the PMF quality parameters and most reasonable source identification. In the seven-factor solution, several factors exhibited mixing across different data segments, whereas solutions with a larger number of factors led to excessive splitting of physically meaningful factors. The mass spectra of the 8 thermograms factors (referred as thermogrAMS-OA factors) of these three data sets can be found in Fig. S7. Since the entire campaign data set was divided into three parts, it is essential to perform the correlation analysis of mass spectra of 8 factors across different data sets (part 1 to 3) to identify the similar factors among the three data sets. In part 2 and 3 data sets, there were clear correlations between the respective factors, suggesting that the PMF results of part 2 and 3 data sets can be reasonably combined.

In part 1 data set, both factors 1 and 6 showed the highest correlation with factor 6 in the part 2 and 3 data set, respectively (fig. S8). However, there are no factors strongly correlated with F1 and F7 in part 2 and 3, respectively. It could be owing to that the sources of OA during 2 to 15 October (part 1) were different from those during 16 October to 16 November (part 2 and 3). In the discussion in sections 3.1 and 3.2, F1 and F7 in part 2 and 3 were believed to originate from photochemical reactions in the urban plumes and biomass burning, respectively. Figure S9 demonstrates that the site was mainly affected by south wind with a relatively lower concentration of BBOA and levoglucosan from 2 to 5 October. Thus, we performed PMF analysis to a new dataset (part 1.5), from 5 to 22 October. Clear correlations between the respective factors were found in part 1.5, 2, and 3 data sets (Fig. S8). Finally, we combined these three data sets (5 October to 16 November) in the manuscript."

[Figure]

**Figure S3.** Diagnostic plots for part 1 (2 to 15 Oct.).

[Figure]

**Figure S4.** Diagnostic plots for part 1.5 (5 to 22 Oct.).

[Figure]

**Figure S5.** Diagnostic plots for part 2 (16 to 30 Oct.).

[Figure]

**Figure S6.** Diagnostic plots for part 3 (31 Oct. to 16 Nov.).

[Figure]

**Figure S7.** The mass spectra of 8 thermograms PMF factor of four data sets (part 1 to 3).

[Figure]

**Figure S8.** The correlation of mass spectra of 8 factors across different data sets (part 1 to 3). The red squares represent the highest R value for a specific factor along the x-axis compared to all factors on the y-axis.

[Figure]

**Figure S9.** The time series of (a) levoglucosan in the particle phase measured by the FIGAERO-CIMS and BBOA in AMS-OA, (b) NO$_x$, and (c) wind speed and direction from 2 October and 16 November.

3. In line 194, the justification for selecting the 8-factor solution is insufficient. Standard PMF diagnostics (Q/Qexp, residuals, Fpeak sensitivity) should be provided.

Reply: We appreciate the reviewer for these valuable suggestions. We have expanded the PMF diagnostics to explicitly justify the factor selection. The evolution of $Q/Q_{exp}$ values with increasing factor numbers, Fpeak sensitivity, and residuals is now described in the main text and shown in the Supplementary Information (Figs. S3–S6), indicating a clear improvement from 2 to 8 factors and more moderate changes thereafter. The stability of the 8-factor solution was evaluated through cross-segment factor comparisons and solution stability tests, as described in response to comment 2. These diagnostics collectively support the selection of the 8-factor solution. Relevant descriptions and references to standard PMF diagnostics have been added to the manuscript in line 194-195,

**"An eight-factor solution was selected for each part based on $Q/Q_{exp}$ behavior and factor interpretability (Fig. S3 to S6)."**

4. In lines 214-220, the PEG-based calibration (PEG 5–8) may not be representative of nitrogen-containing or highly oxygenated organic species. Calibration uncertainties should also be discussed.

Reply: We appreciate the reviewer for this valuable suggestion. We acknowledge that that the volatility of PEG 5-8 (-1.73 $\leq \log_{10} C^* \leq$ 3.34 µg m$^{-3}$) might not be able to cover a volatility range of nitrogen-containing or highly oxygenated organic species, which usually had a lower volatility (Ren et al., 2022). Unfortunately, currently available saturation vapor pressure data for PEG standards only extend up to PEG-8 (Krieger et al., 2018). Ylisirniö et al. (2021) demonstrated that different extrapolation approaches for estimating the volatility of higher-order PEGs can lead to substantial discrepancies in calibration results, and they strongly recommended that higher-order PEGs should only be used to extend the volatility calibration range once their saturation vapor pressures are accurately determined. Very recently, Ylisirniö et al. (2025) derived saturation vapor pressures for higher-order PEGs up to PEG-15 and demonstrated that extending FIGAERO-CIMS calibration to much lower volatilities is feasible, but also showed that different estimation approaches for higher-order PEGs can lead to large discrepancies, highlighting substantial uncertainties when extrapolating volatility calibration beyond PEG-8.

We added some discussion in the revised manuscript in line 232-248,

**"It was worth noting that the volatility range of PEG 5-8 (-1.73 $\leq \log_{10} C^* \leq$ 3.34 µg m$^{-3}$) may not fully represent the volatility of ambient organic aerosol, particularly nitrogen-containing and highly oxygenated compounds that can**

exhibit much lower volatility ($\log_{10} C^* \leq -2$ µg m$^{-3}$) (Ren et al., 2022; Chen et al., 2024). At present, saturation vapor pressure data for PEG standards are only available up to PEG-8 (Krieger et al., 2018). Ylisirniö et al. (2021) demonstrated that different extrapolation approaches for estimating the volatility of higher-order PEGs can lead to substantial discrepancies in calibration results, and they strongly recommended that higher-order PEGs should only be used to extend the volatility calibration range once their saturation vapor pressures are accurately determined. Very recently, Ylisirniö et al. (2025) derived saturation vapor pressures for higher-order PEGs up to PEG-15 and demonstrated that extending FIGAERO-CIMS calibration to much lower volatilities is feasible, but also showed that different estimation approaches for higher-order PEGs can lead to large discrepancies, highlighting substantial uncertainties when extrapolating volatility calibration beyond PEG-8. Therefore, uncertainties may remain in the calibration of low-volatility OA, and further calibration experiments using complementary techniques are highly recommended.”

5. In lines 257-260, the manuscript acknowledges decomposition artifacts for some species (e.g., C2–C3), but does not systematically address pyrolysis across all factors. A more comprehensive evaluation is required.

Reply: We appreciate the reviewer for this valuable suggestion. The C2–C3 groups showed significant contributions only in the Day-LNOx-LVOA and Day-aged-ELVOA factors (Fig. 1). We further investigate the contribution of FIGAERO factors to the signal of $C_2H_2O_3$, $C_2H_4O_3$, $C_3H_4O_3$, and $C_3H_6O_3$. The results indicate that Day-aged-ELVOA made a non-negligible contribution to all four species, especially for $C_2H_4O_3$ and $C_3H_6O_3$.

The thermogram of $C_2H_2O_3$ and $C_3H_4O_3$ exhibited a bimodal distribution: one mode peaking in the LVOC range, which was mainly associated with Day-LNO$_x$-LVOA, and a second mode peaking in the ELVOC range, dominated by Day-aged-ELVOA. Contributions from other factors were comparatively minor. These results suggest that the thermal desorption behavior of these C$_2$–C$_3$ species can be largely explained by the combined influences of Day-LNO$_x$-LVOA and Day-aged-ELVOA.

We add some discussion in the revised manuscript in line 278-285,

“Noting that C2-C3 group could originate from the decomposition of larger molecules during thermal desorption, since the thermogram of $C_2H_2O_3$ and $C_3H_4O_3$ demonstrated a bimodal distribution (Fig. 9 a). Figure S9 b and d further examine the contribution of all FIGAERO factors to the signals of $C_2H_2O_3$ and $C_3H_4O_3$. One mode, peaking in the

LVOC range, was primarily associated with Day-LNO$_x$-LVOA, and a second mode, peaking in the ELVOC range, was dominated by Day-aged-ELVOA. These results indicates that these two low molecular weight species are likely decomposition products of at least two distinct classes of higher molecular weight organic compounds."

We have also revised the sentence previously located at lines 302-308 to:

**"However, C$_2$H$_4$O$_3$ and C$_3$H$_6$O$_3$ had a weak correlation (R=0.49 and 0.13) with MO-OOA resolved from AMS (Fig. S11). In addition, the $T_{max}$ of C$_2$H$_4$O$_3$ and C$_3$H$_6$O$_3$ located in the ELVOC range and showed thermogram profiles similar to that of Day-aged-ELVOA (Fig. S12a). The thermogram signal of C$_2$H$_4$O$_3$ and C$_3$H$_6$O$_3$ was mainly contributed by Day-aged-ELVOA (Fig. S12 c and e), supporting the interpretation that these species are more likely decomposition products of low volatility organic compounds rather than being directly formed through atmospheric aging processes."**

6. The chemical characteristics of Day-urban-LVOA and Day-HNOx-LVOA overlap significantly. More evidence is needed to show they are not artifacts of factor splitting. (in lines 275-280 and Table 1)

Reply: We appreciate the reviewer for this valuable suggestion. We acknowledge that the volatility, H:C, and O:C of these two factors are similar. However, the oxidation state ($\overline{OS_c}$) of Day-HNO$_x$-LVOA (-0.01) was significantly lower than that of Urban-LVOA (0.08), accompanied by a relatively higher N:C (0.06 vs 0.04). Despite its lower oxidation state, the volatility of Day-HNO$_x$-LVOA is comparable to that of Day-urban-LVOA, likely due to its higher nitrogen content. Organic nitrates are known to have lower volatility than hydroxylated products with the same carbon number (Donahue et al., 2011; Ren et al., 2022).

We further investigated the temporal variations of these two factors and found that Day-urban-LVOA showed only a limited similarity in its variation trend to Day-HNOx-LVOA during the urban air mass period. This behavior suggests that Day-HNO$_x$-LVOA and Day-urban-LVOA are formed through distinct atmospheric pathways.

We add some discussion in the revies manuscript in line 313-318,

**"However, the oxidation state ($\overline{OS_c}$) of Day-HNO$_x$-LVOA (-0.01) was significantly lower than that of Urban-LVOA (0.08), accompanied by a relatively higher N:C (0.06 vs 0.04). Despite its lower oxidation state, the volatility of Day-HNO$_x$-LVOA is comparable to that of Day-urban-LVOA, likely due to its higher nitrogen content. Organic nitrates generally exhibit lower volatility than hydroxylated products with the same carbon number (Donahue et al., 2011; Ren et al., 2022)."**

Additional discussion added in Section 3.2 in line 503-504,

**"In addition, Day-urban-LVOA showed only a limited similarity in its variation trend to Day-HNOx-LVOA during the urban air mass period (Fig. S26)."**

[Figure]

**Figure S26.** Temporal variation of Day-HNO$_x$-LVOA and Day-urban-LVOA.

7. In lines 314-316, the deviation of Day-urban-ELVOA from the expected relationship is attributed simply to "decomposition". This may require a more rigorous discussion.

Reply: We appreciate the reviewer for this valuable suggestion. We further investigate the thermogram of the major organic molecules ($C_5H_6O_4$, $C_4H_6O_5$, $C_6H_8O_4$, and $C_8H_{10}O_5$), as well as their contribution from all FIGAERO factors. The results show that these molecules do not exhibit thermograms like that of Day-urban-ELVOA. Instead, their thermograms demonstrate multimodal distributions and are contributed by multiple FIGAERO factors.

For example, a mode of $C_5H_6O_4$ peaking in the LVOC range was mainly contributed by Day-urban-LVOA, while two modes peaking in the ELVOC range were primarily contributed by Day-aged-ELVOA and Day-urban-ELVOA, respectively. These results suggest that these molecules may originate from both direct desorption of organic aerosol and thermal decomposition of higher-molecular-weight compounds during heating.

We added some discussion in the revised manuscript in line 322-330,

**"However, the majority of organic molecules (e.g., $C_5H_6O_4$, $C_4H_6O_5$, $C_6H_8O_4$, and $C_8H_{10}O_5$) do not exhibit thermograms similar to that of Day-urban-ELVOA (Fig. S13). Instead, their thermograms demonstrate multimodal distributions and are contributed by multiple FIGAERO factors. For example, a mode of $C_5H_6O_4$**

**peaking in the LVOC range was mainly contributed by Day-urban-LVOA, while two modes peaking in the ELVOC range were primarily contributed by Day-aged-ELVOA and Day-urban-ELVOA, respectively. These results suggest that these molecules may originate from both direct desorption of organic aerosol and thermal decomposition of higher-molecular-weight compounds during heating.**"

[Figure]

**Figure S13.** (a) The average thermogram of $C_5H_6O_4$, $C_4H_6O_5$, $C_6H_8O_4$, and $C_8H_{10}O_5$; (b-e) The thermogram signal of each ion contributed by all FIGAERO factors.

8.  In lines 386-418, the interpretation of NOx effects is speculative without supporting evidence from highly oxygenated organic molecules or accretion reaction markers. Although the manuscript proposes that NOx suppresses autoxidation and shifts SOA formation toward more volatile and less oxygenated components, this conclusion is currently based primarily on correlations and factor behavior. To substantiate this mechanism, molecular-level evidence would be necessary. The authors should

therefore adopt more cautious wording or provide additional analyses to better support their proposed NOx-driven interpretation.

Reply: We appreciate the reviewer for this valuable suggestion. We acknowledge that the quantification of larger multifunctional organic species, including potential accretion (dimer) products, by I-CIMS is inherently uncertain due to highly variable instrument sensitivity to different molecular structures and the lack of calibration standards. (Lee et al., 2014). Bi et al. (2021) further demonstrated that sensitivities of different isomers with the same elemental formula measured by iodide-CIMS can vary by up to two orders of magnitude, and sensitivity predictions using voltage scanning also carry high uncertainties for individual analytes. This implies that without specific calibration, quantification of complex oxidation products, including potential oligomers or accretion products, by I-CIMS may be inherently uncertain.

To provide molecular level evidence, we investigate diurnal evolution of organic compositions under long-range transport period (low $NO_x$) and urban air masses (high $NO_x$) period (Fig. S22). Mass concentrations of CHON increase during the daytime in both periods, with a more pronounced enhancement observed in urban air masses (Fig. S22a). However, the mass fraction of CHON was lower during the urban air masses period than during the long-range transport period. We speculated that elevated $NO_x$ enhances overall oxidation and product formation rather than selectively enriching nitrogen-containing compounds. This interpretation is consistent with results from our previous observation-constrained box-model simulations, in which production rates of OH and organic peroxy radicals ($RO_2$) were evaluated under varying NOx and VOC conditions (Cai et al., 2024). The modeled $P$(OH) were close to the transition regime, indicating that elevated $NO_x$ can enhance atmospheric oxidation capacity. In contrast, the $P$($RO_2$) was in the VOC-limited regime and decreased with increasing $NO_x$.

Consistent with these results, Fig. S22c shows that the mass fraction of highly oxygenated organic molecules ($O \geqslant 6$) is lower during urban air masses period. Concurrently, species with low oxygen numbers ($O \leqslant 3$) become relatively more abundant in the urban plumes (Fig. S22c), indicating a shift in the oxidation product distribution toward less oxygenated and potentially more volatile compounds, the $NO_x$-driven suppression of multigenerational autoxidation inferred from the box-model results. This suppression of oxidation is observed for both CHON and CHO species. The average O:C of CHON (Fig. S22b) and CHO (Fig. S22e) are both lower during the urban air masses period, suggesting that enhanced $NO_x$ broadly suppresses autoxidation across organic compound classes.

We added some discussion in the revised manuscript in line 459-478,

"We investigate diurnal evolution of organic compositions under long-range transport and urban air masses periods (Fig. S22). Mass concentrations of CHON increase during the daytime in both periods, with a more pronounced enhancement observed in urban air masses (Fig. S22a). However, the mass fraction of CHON was lower during the urban air masses period than during the long-range transport period. We speculated that elevated $NO_x$ enhances overall oxidation and product formation rather than selectively enriching nitrogen-containing compounds. This interpretation is consistent with results from our previous observation-constrained box-model simulations, in which production rates of OH and organic peroxyl radicals ($RO_2$) were evaluated under varying NOx and VOC conditions (Cai et al., 2024). The modeled $P$(OH) were close to the transition regime, indicating that elevated $NO_x$ can enhance atmospheric oxidation capacity. In contrast, the $P$($RO_2$) was in the VOC-limited regime and decreased with increasing $NO_x$. Consistent with these results, Fig. S22c shows that the mass fraction of highly oxygenated organic molecules (O≥6) is lower during urban air masses period. Concurrently, species with low oxygen numbers (O≤3) become relatively more abundant in the urban plumes (Fig. S22c), indicating a shift in the oxidation product distribution toward less oxygenated and potentially more volatile compounds, the $NO_x$-driven suppression of multigenerational autoxidation inferred from the box-model results. This suppression of oxidation is observed for both CHON and CHO species. The average O:C of CHON (Fig. S22b) and CHO (Fig. S22e) are both lower during the urban air masses period, suggesting that enhanced $NO_x$ broadly suppresses autoxidation across organic compound classes."

[Figure]

**Figure S22.** Diurnal variations of (a) mass concentration of CHON compounds, (b) the average O/C of CHON, (c) the mass fraction of highly oxygenated species (O ≥ 6), (d) the mass fraction of

CHON, (e) the average O/C of CHO, and (f) the mass fraction of low-oxygen species during different periods.

**Reference:**

Bi, C., Krechmer, J. E., Frazier, G. O., Xu, W., Lambe, A. T., Claflin, M. S., Lerner, B. M., Jayne, J. T., Worsnop, D. R., Canagaratna, M. R., and Isaacman-VanWertz, G.: Quantification of isomer-resolved iodide chemical ionization mass spectrometry sensitivity and uncertainty using a voltage-scanning approach, Atmos. Meas. Tech., 14, 6835-6850, 10.5194/amt-14-6835-2021, 2021.

[revised manuscript text omitted]

---

## Author Response (AR2)

**Editor**

Dear authors,

Thank you for your detailed and thoughtful response to the reviewers' comments. I have one remaining minor comment/question and would like to ask for a brief revision before the manuscript is formally accepted.

In both the response document and the main text, the statement appears multiple times that "organic nitrates generally exhibit lower volatility than hydroxylated products with the same carbon number." In my view, this statement is somewhat ambiguous. Within commonly used frameworks such as VBS or SIMPOL, a nitrate group (-ONO2) contributes to volatility reduction at a level comparable to a hydroxyl group (-OH), while the -NO2 moiety itself does not substantially reduce volatility. It is therefore unclear whether your observation reflects systematically higher oxidation states of the organic nitrates, differences in O-containing functional groups, or some other factor.

I therefore ask that you please revisit this statement and revise the relevant sections of the manuscript, as appropriate, to ensure that the description of functional group effects on volatility is accurate and clearly conveyed.

Reply: We appreciate the editor for this valuable suggestion. In this statement, we aimed to discuss the reason why Day-HNO$_x$-LVOA exhibits a volatility comparable to that of Day-urban-LVOA, despite their substantially different oxidation states (-0.01 vs 0.8). We acknowledge that attributing this behavior solely to the presence of organic nitrates is ambiguous, as variations in volatility may arise from the combined effects of multiple functional groups rather than a single functional group

However, given the limitations of our measurement techniques, it is challenging to directly identify the specific functional groups associated with different OA factors. Thus, we revised this sentence in the former line 315-318 as follows,

**"Despite its lower oxidation state, the volatility of Day-HNOx-LVOA is comparable to that of Day-urban-LVOA, which may reflect differences in functional group composition. For example, a nitrate group (-ONO$_2$) contributes to volatility reduction at a level comparable to that of a hydroxyl group (-OH) and generally more strongly than carbonyl functionalities such as aldehydes (–C(O)H) or ketones (–C(O)–) (Pankow and Asher, 2008). However, due to instrumental limitations, we are unable to directly resolve the functional group composition of individual OA factors, and further measurements employing new**

**techniques are needed to better constrain the role of functional groups in controlling the volatility of ambient organic aerosol**."

In addition, we have deleted the sentence in the former lines 593–595,

**"Moreover, organic nitrates generally have lower volatility than hydroxylated species with the same carbon number (Donahue et al., 2011; Ren et al., 2022). It suggested that a higher fraction of nighttime organic nitrates could lead to lower OA volatility (Kiendler-Scharr et al., 2016)."**

**Reference:**

Pankow, J. F. and Asher, W. E.: SIMPOL.1: a simple group contribution method for predicting vapor pressures and enthalpies of vaporization of multifunctional organic compounds, Atmos. Chem. Phys., 8, 2773-2796, 10.5194/acp-8-2773-2008, 2008.